# Mid-infrared light resonance-enhanced proton conductivity in ceramics

Haobo Li [1], Yicheng Zhu [1], Zihan Zhao[1], Ruixin Ma[1], Jiachen Lu [1], Wenjie Wan [1,2] ✉ & Qianli Chen [1] ✉

Ionic transport in solids is a critical process for energy devices including batteries and fuel cells. To improve ionic transport, an emerging approach is the selective excitation of atomic vibrations related to the mobile ions. However, there is limited direct experimental evidence demonstrating enhanced macroscopic ionic conductivity through this approach. Here, we use a 140 mW continuous-wave mid-infrared (MIR) light to excite the O−H stretch vibration in proton-conducting yttrium-doped barium zirconate. We observe reversible enhancement of 36.8% in bulk, and 53.0% in grain boundary proton conductivities, controlled by MIR irradiation. Decreases in the activation energy and prefactor for bulk proton conduction suggest possible reduction in activation entropy and attempt frequency of proton hopping. We rationalize the enhancement as the excitation of O−H stretch vibrational states, followed by the relaxation into lattice vibration modes, modulating the potential energy surface of the proton. Our findings highlight MIR irradiation as a power-saving strategy to optimize the performance and operation cost of solid-state electrochemical devices by selective modulation of the vibrational properties.

Fast ionic conductors are essential materials in various electrochemical devices such as solid-state batteries[1,2], fuel cells[3,4], electrolysers[5,6], and gas sensors[2,7]. These devices use pure ionic conductors as electrolytes[4,8] and mixed ionic-electronic conductors as electrodes[9,10]. Fast ionic conduction in a variety of materials has been investigated, ranging from lithium and sodium ion conductors including $Li_{0.5}La_{0.5}TiO_3$, $Li_{10}GeP_2S_{12}$, $Na_3SbS_4$, and $\beta$-alumina[11,12], to proton and oxide ion conductors including acceptor-doped $Ba(Zr,Ce)O_3$, $Zr_{1-x}Y_xO_2$, and $Ce_{1-x}Gd_xO_2$[13,14]. Enhancing the ionic conductivity remains a critical challenge for the material design of high-performance fast ionic conductors[14,15]. Researchers have dedicated their efforts to improving the ionic conductivity by tuning the mobile ion concentration[16–18], material structure and composition[4,19,20], and processing routes[21–23].

Despite various types of mobile ions, ionic transport in fast ionic conductors is classically described by the hopping mechanism, of ion hops from one site to an adjacent vacancy or interstitial site by overcoming an energy barrier[15,24]. Atomic vibrations, including lattice and localised vibrations, play an important role in ionic transport[25]. Lattice vibrations (phonons) have been theoretically suggested to provide pathways for ionic hopping[26–28]. Moreover, several studies proposed that large-amplitude localised vibration of O−H and oxygen vacancies can induce proton and oxide ion hopping in solids[14,29]. For instance, proton transport in acceptor-doped $BaZrO_3$ occurs through sequential hopping via O−H bond breaking and formation, and the O−H stretch vibration is reported to strongly impact proton hopping[14,24].

Selective excitation of atomic vibrations related to the mobile ions provides a promising tool to promote ionic conduction. This strategy usually requires terahertz[30–32] or infrared radiation[33–37] or Raman pumping[38,39] that resonate with the targeted vibrations. In several pioneering works, the atomic vibrations contributing to the majority of ion hopping energy in some solid-state ionic conductors have been identified and excited to enhance ionic conductivity. Gordiz et al.[40], Pham et al.[41,42] and Poletayev et al.[43] have demonstrated the potential of selectively exciting lattice vibrations in the terahertz range to promote ion conduction in $Li^+$ conductors $Li_3PO_4$, $Li_{0.5}La_{0.5}TiO_3$, and $Na^+$-, $K^+$-, $Ag^+$-conducting $\beta$-aluminas. Spahr et al. resonantly excited

[1]Global College, Shanghai Jiao Tong University, Shanghai 200240, China. [2]School of Physics and Astronomy, Shanghai Jiao Tong University, Shanghai 200240, China. ✉e-mail: wenjie.wan@sjtu.edu.cn; qianli.chen@sjtu.edu.cn

the O−H stretch vibration in proton-conducting single-crystal KTaO$_3$[35] and rutile TiO$_2$[36] using mid-infrared (MIR) laser at room temperature and achieved giant enhancement of microscopic proton hopping rates by 7 and 9 orders of magnitude, respectively. However, direct experimental evidence of enhancing macroscopic ionic conductivity through selective excitation of atomic vibrations is still rare[41,42].

Light has been used as a stimulation source to enhance the ionic conductivity of solids via other mechanisms. Defferriere et al. enhanced oxide ion conductivity across the grain boundaries in Gd-doped ceria films by illumination with above-bandgap light, and suggested that the photogenerated electrons reduced the potential barrier for ionic conduction in the space-charge regions of grain boundaries[44]. In coordination polymer glass, Ma et al. reported switchable proton conductivity modulated by ultraviolet-visible light, which generates proton deficient sites and creates low-barrier proton transfer paths in the glass[45].

In this work, we report the enhanced proton conductivity by exciting the O−H stretch vibration using continuous-wave (CW) MIR light in protonated BaZr$_{0.8}$Y$_{0.2}$O$_{3-\delta}$ (BZY), a representative proton conducting oxide. The principle of our experiment is illustrated in Fig. 1. BZY is an ideal model system for our investigation, in particular, because the frequency of O−H stretch vibration is well-separated from those of lattice vibrations, allowing investigations on specific vibration modes. Moreover, the O−H stretch vibration wavelength of ~3 μm enables resonant excitation using readily accessible MIR light sources[46–48]. The 3 μm wavelength corresponds to a rather high vibrational excitation energy (~0.4 eV), comparable to the activation energy for bulk proton conduction in BZY[20,49]. Therefore, it is reasonable to believe that selective excitation of O−H stretch vibration can provide energy to facilitate proton hopping, and promote proton conduction.

We demonstrate reversible switching between high and low bulk and grain boundary (GB) resistances controlled by MIR irradiation measured by electrochemical impedance spectroscopy (EIS) while maintaining the sample temperature constant. The enhancement effect in proton conductivity is 2-3 times greater than the estimated IR heating effect. Bandpass filtering was further applied to highlight the enhancement effects of MIR wavelength resonant to the O−H stretch frequency. We rationalize our observation by modelling the excitation of the O−H stretch vibration in the potential energy surface (PES) of the proton.

## Results and discussion

Polycrystalline BZY pellets (inset of Fig. 1c) with a thickness of ~0.4 mm were prepared by the solid-state reaction method[27,50]. The pellet thickness was chosen to ensure effective MIR penetration (Supplementary Fig. 17). The protonated pellets painted with silver (Ag) paste-coated electrodes were mounted on a lab-made test system for MIR irradiation and conductivity measurements. The pellets were firmly fixed to the heating plate to ensure uniform heating. To minimize the heating effect induced by MIR irradiation, a thermostat with a feedback loop was employed in the system to maintain the sample at a constant temperature. As illustrated in Fig. 2a, the samples were irradiated by a CW MIR light with an output power of 140 mW. Their electrical conductivities were measured by EIS with and without MIR irradiation in H$_2$O-saturated N$_2$ (wet N$_2$; $p$(H$_2$O) = 0.02 atm) between 130 and 200 °C. This temperature range was chosen since both bulk and GB responses can be identified in the EIS spectra for BZY[51]. The O−H stretch vibration of BZY appears as a broad band between 2.70 and 4.00 μm[47,48], as characterized by diffuse reflectance infrared Fourier transform spectroscopy (DRIFTS) shown in Fig. 2b. The emission spectrum of the MIR source spans from 2 μm−6 μm, covering the absorption spectrum of O−H stretch vibration. Details for sample preparation, characterization, and the lab-made test system are provided in Methods and Supporting Information.

### Enhanced proton conductivity upon MIR irradiation in BZY

Figure 2c shows representative EIS spectra at 160 °C with and without MIR irradiation. The corresponding effective MIR power density ($p$), defined as the output power density of the MIR light within the wavelength range of O−H stretch vibration, is 195.2 mW cm$^{-2}$. Figure 2c and Supplementary Fig. 12, 13 indicate that the real part of impedance ($Z'$) at 150 kHz and 1 kHz reflect bulk and GB proton conduction features in the EIS spectra. The single-frequency impedance at these two frequencies shows that MIR irradiation resulted in an immediate drop in $Z'$ for both features (Fig. 2d, e). When the MIR light was switched on, a small fluctuation of $Z'$ was observed within the first 100 seconds, as a result of temperature modulation by the thermostat to compensate for the heat induced by MIR irradiation. After around 100 s, $Z'$ reached stable values. Upon removal of MIR irradiation, $Z'$ returned to their original values, confirming the reversibility of the enhancement effect on proton conduction through MIR irradiation. Figure 2f compares the proton conductivities after the MIR light was switched on or off for 100 s.

The EIS spectra in Fig. 2c were then fitted to an equivalent circuit consisting of three serial R(CPE) elements, assuming proton conduction as the origin of the impedance. Each R(CPE) element represents the contribution from the bulk, GB, and electrode process to the overall impedance. From the fitting parameters of the EIS spectra, an obvious decrease in both bulk and GB resistances ($R$) under MIR irradiation was revealed, as presented in Fig. 2f. The enhancement in

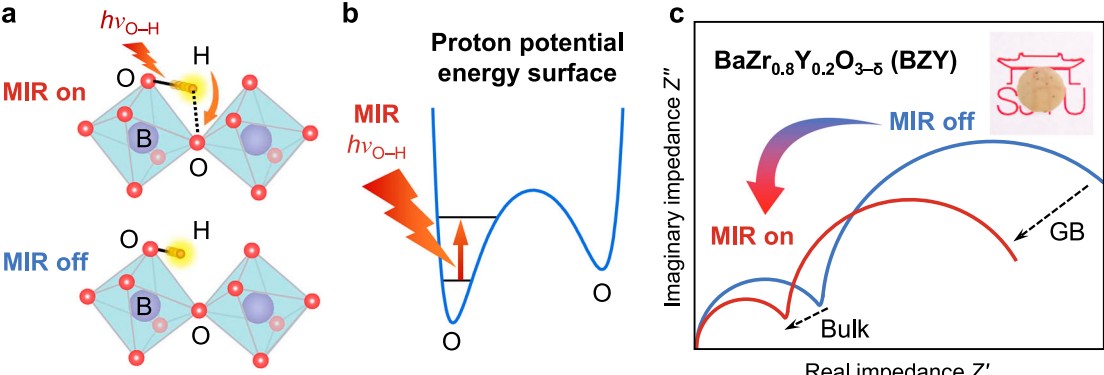

**Fig. 1 | Principle of MIR light enhanced proton conductivity through excitation of O−H vibration. a** Upon resonant MIR excitation with the energy of $h\nu_{O-H}$, the O−H stretch amplitude is increased to facilitate proton transfer. **b** MIR light excites the O−H stretch vibration from the ground state to an excited state. **c** Schematics of the corresponding EIS spectra with (red) and without (blue) MIR irradiation. Inset: photo of a 0.4-mm thick protonated polycrystalline BZY pellet. In (**a**) the purple, red and orange spheres represent B-site (Zr or Y), O and H atoms respectively, while the light blue squares denote BO$_6$ (B = Zr, Y) octahedra.

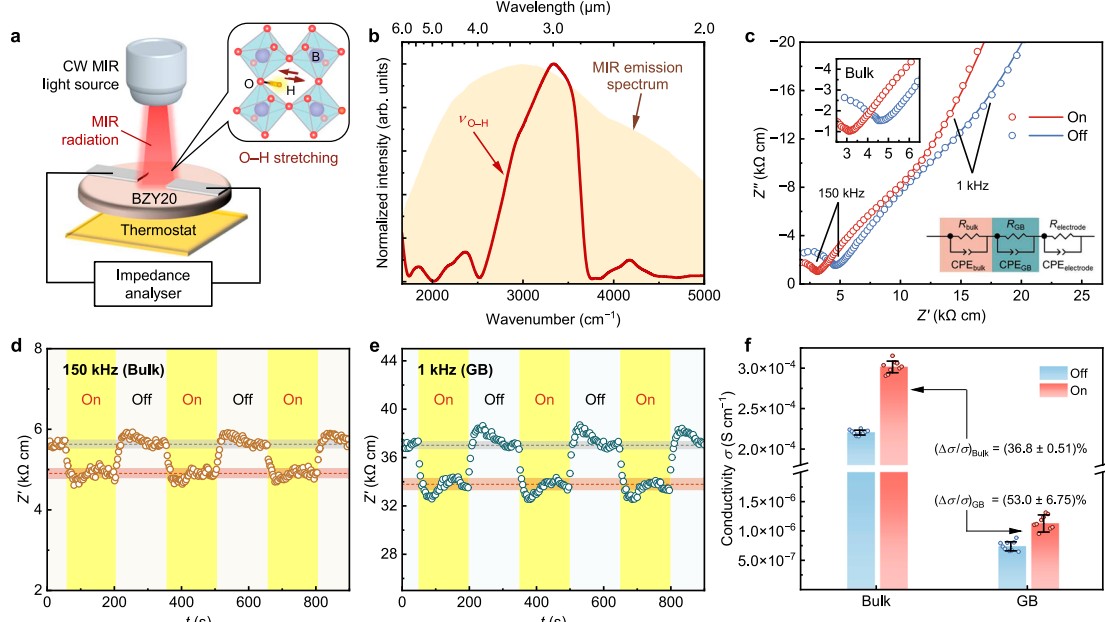

**Fig. 2 | Characteristics of O–H stretch vibration and proton conduction with and without MIR irradiation. a** Schematics of the electrical conductivity measurement setup. The MIR radiation covers the wavelength range of O–H stretch vibration. **b** Vibrational profile of O–H stretch vibration band ($\nu_{O-H}$) characterized by DRIFTS (red). The orange shadow shows the typical emission spectrum of the CW MIR light. **c** Typical EIS spectra at 160 °C in wet $N_2$ with (red) and without (blue) MIR iOrradiation at an effective MIR power density ($p$) of 195.2 mW cm$^{-2}$. The insets illustrate the magnification of bulk features near 150 kHz, and the equivalent circuit used for fitting. The reversibility of proton conductivity enhancement is represented by single-frequency $Z'$ at (**d**) 150 kHz for bulk (brown) and (**e**) 1 kHz for GB (turquoise). The yellow shadows highlight the time intervals for MIR irradiation. The dashed red and grey lines show steady-state $Z'$ averages with and without irradiation respectively; error bands indicate the corresponding standard deviations. **f** Comparison of bulk and GB proton conductivities deconvoluted from **c** and corresponding enhancement ratio of conductivity ($\Delta\sigma/\sigma$). The error bars in (**f**) incorporate standard deviations from three parallel measurements and fitting errors per sample across three samples prepared using the same method. Source data are provided in the Source Data file.

proton conductivity under MIR irradiation is then quantified with the enhancement ratio or percent change of conductivity ($\Delta\sigma/\sigma$):

$$\frac{\Delta\sigma}{\sigma} = \left(\frac{\sigma^{on}}{\sigma^{off}} - 1\right) \times 100\% = \left(\frac{R^{off}}{R^{on}} - 1\right) \times 100\% \quad (1)$$

where $\sigma^{on}$, $R^{on}$, and $\sigma^{off}$, $R^{off}$ denote the conductivity and resistance, with and without MIR irradiation, respectively. The enhancement ratios of bulk and GB conductivities, denoted as $(\Delta\sigma/\sigma)_{Bulk}$ and $(\Delta\sigma/\sigma)_{GB}$, are as high as 36.8% and 53.0% (Fig. 2f). The GB conductivity ($\sigma_{GB}$) is defined as the specific value derived from the brick-layer model (Supplementary Note 6) to account for the microstructure[52,53]. Additionally, no significant changes were observed in the EIS spectra measured in wet $N_2$, ambient air, and dry $N_2$ atmospheres (Supplementary Fig. 5), indicating a negligible atmosphere effect at 160 °C. The changes in impedance were not significant when changing water partial pressure either with or without MIR irradiation, suggesting that the conductivity is not a surface protonic effect[54]. Nevertheless, the enhancement was prominent when the MIR light was switched on, confirming that the measured enhancement in electrical conductivity originates from bulk transport.

To explore the potential of enhancement in proton conductivity through MIR excitation in BZY, $\Delta\sigma/\sigma$ was evaluated versus various $p$ by adjusting the distance ($d$) between the sample and MIR light source. $p$ was calibrated as a decreasing function of $d$ (Supplementary Fig. 10b). $\Delta\sigma/\sigma$ for both bulk and GB were found to linearly scale with $p$ (Fig. 3a, b), suggesting that $\Delta\sigma/\sigma$ is linearly correlated with the number of MIR photons incident onto the samples. The intercepts of both lines with the $x$-axis are >0, which may arise from the absorption of MIR radiation by $H_2O$ molecules in humid atmosphere. To the best of our knowledge, the currently available $p$ for MIR laser working in CW mode can be up to 16 W cm$^{-2}$[55], and such power density is also achievable

with an IR furnace[34,56]. Then, $\sigma^{on}/\sigma^{off}$ for both bulk and GB could be further enhanced by 30–50 times. Consequently, the bulk and GB conductivities are expected to be above $10^{-3}$ S cm$^{-1}$ and $10^{-5}$ S cm$^{-1}$, respectively. Such enhancement in conductivity is equivalent to heating the sample to 335 °C for the bulk and 275 °C for the GB, while maintaining sample temperature at 160 °C. Therefore, it is possible to achieve an over-ten-fold enhancement in proton conductivity through MIR irradiation.

IR irradiation is known to induce heat, which may also cause enhancement in the ionic conductivity. Since our measurement setup cannot completely exclude the IR heating effect, here, we distinguish the contribution of heat to the observed enhancement in conductivity through an estimation of $\Delta\sigma/\sigma$ by IR heating effect. It is assumed that the samples absorb all spectral components radiated from the MIR light source, and all the incident optical power is directly converted into heat. The consequent increase in temperature is estimated by considering the sample as a thermal resistor for thermal conduction[44], as elaborated in Supplementary Note 8. Depending on different values of $d$ (corresponding to different power densities for heating), the increase in sample temperature is within 1–5 °C, and the corresponding results of $\Delta\sigma/\sigma$ are plotted in Fig. 3c, d. The observed $\Delta\sigma/\sigma$ for both bulk and GB due to MIR irradiation are about 2 times greater than the estimated results due to the IR heating effect. Therefore, IR heating effect is not a major contributor to the enhancement in proton conductivity. Instead, the excitation of the O–H stretch vibration by absorbing the MIR radiation is considered to dominate the enhancement in proton conductivity. To reach the same proton conductivity, the temperature rise and thermal expansion due to IR heating is still less than the effect caused by thermal heating (Supplementary Notes 8–9).

To confirm that the observed effect originates from enhanced proton conductivity, we further conducted the experiments on

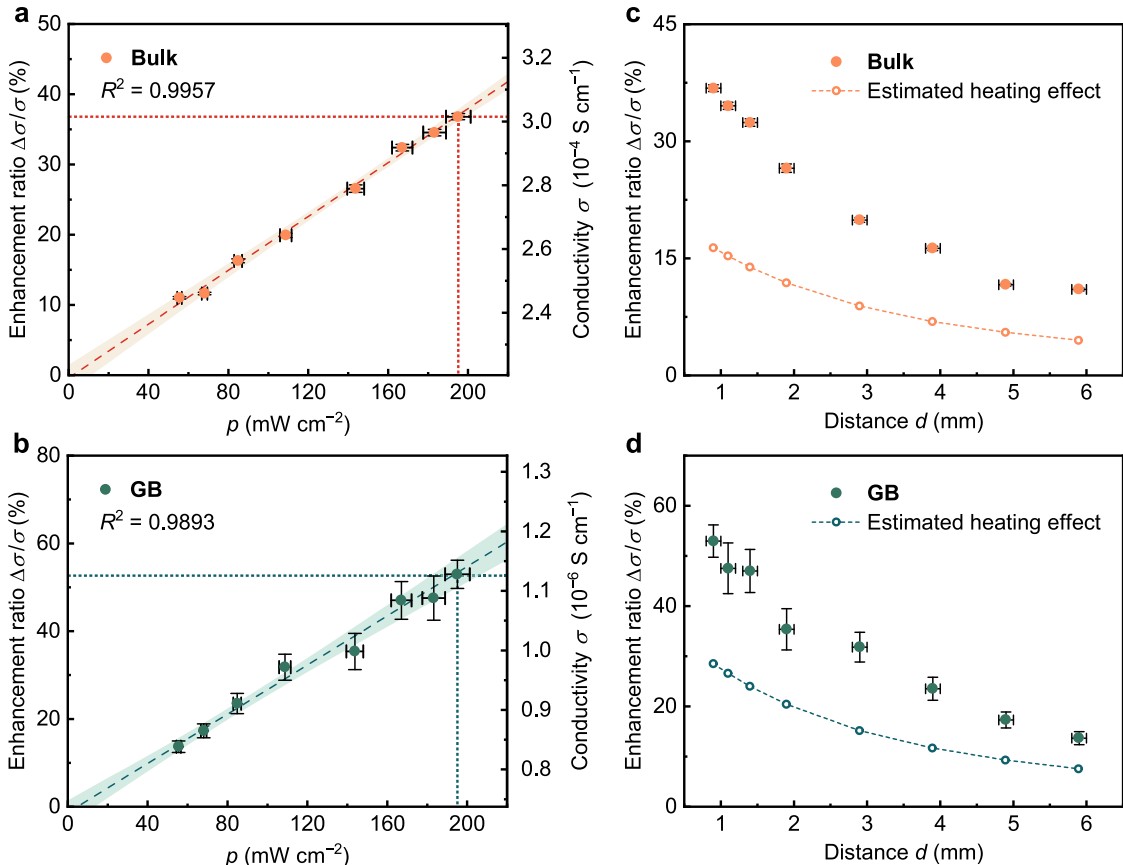

**Fig. 3 | Enhancement in proton conductivity versus MIR irradiation intensity.**
$\Delta\sigma/\sigma$ for (**a**) bulk (orange) and (**b**) GB (turquoise) versus $p$ controlled by adjusting the distance between the sample and MIR light source ($d$). The sample was maintained at 160 °C. Shadowed areas in (**a**) and (**b**) illustrate the 95% confidence band of the fitted curves. Comparison of $\Delta\sigma/\sigma$ for (**c**) bulk and (**d**) GB observed in the experiments (closed circles) and estimated with the radiative heating effect (open circles connected with dashed line). The error bars in (**a**–**d**) incorporate standard deviations from three parallel measurements and fitting errors per sample across three samples prepared using the same method. Source data are provided in the Source Data file.

deuterated samples in $D_2O$-saturated $N_2$ ($p(D_2O) = 0.02$ atm) and dry samples in dry $N_2$ ($p(H_2O) < 10^{-4}$ atm). As shown in Supplementary Table 2, the enhancement ratios of both bulk and GB conductivities for the deuterated samples are 23.1% and 33.6%, approximately 0.6 times those of the protonated samples due to the heavier deuteron. For the dry samples, the enhancement ratios are only 3.91% and 6.54%. These results confirm that the enhancement in conductivity is dominated by contributions specific to protons.

### Analysis of proton conductivity parameters with and without MIR irradiation

To illustrate the mechanism of enhanced proton conductivity under MIR irradiation, parameters for temperature-dependent proton conductivity with and without irradiation are quantified and compared. Using classical hopping theory, the prefactor $\sigma_0$ and activation energy $E_a$ (also known as activation enthalpy) of Grotthuss-type proton conductivity $\sigma$ in BZY are evaluated according to the Arrhenius relation[15,57]:

$$\sigma = \frac{\sigma_0}{T}\exp\left(-\frac{E_a}{k_BT}\right) \propto \frac{1}{T}\nu_0 \exp\left(\frac{\Delta S}{k_B}\right)\exp\left(-\frac{E_a}{k_BT}\right) \quad (2)$$

where $k_B$ is the Boltzmann constant, $T$ is the absolute temperature in Kelvin, $\nu_0$ is the attempt frequency, and $\Delta S$ is the activation entropy arising from the change in vibration frequencies at the initial and transition states during a local hopping process[58]. Eq. (2) shows that $\sigma_0$ is strongly dependent on the entropic term $\exp(\Delta S/k_B)$ and attempt frequency $\nu_0$. Given the thermal activation feature of the proton

hopping process, the hopping frequency $\nu$ is described by:

$$\nu = \nu_0 \exp\left(\frac{\Delta S}{k_B}\right)\exp\left(-\frac{E_a}{k_BT}\right) \quad (3)$$

In practice, the variation trend of $\nu$ upon MIR irradiation can be evaluated from the fitting results of EIS spectra[59]. The equivalent circuit model in Fig. 2c provides the values of resistance ($R$) and pseudocapacitance of CPE ($C$). Specifically, $C = Y^{1/n}R^{(1-n)/n}$, where $Y$ and $n$ are capacitance factor and ideality factor, respectively. Subsequently, we can obtain the characteristic frequency for bulk or GB processes, $\omega$, using $\omega = 1/(RC)$[15,59]. The concerted effect of attempt frequency and entropy on proton conductivity was estimated similarly as a recent study on solid-state lithium-ion conductors by Li et al.[57], using a vibrational factor $Q = \omega_0\exp(\Delta S/k_B) = \omega\exp(E_a/k_BT)$, as derived from Eq. (3). Although $\omega$ and $\omega_0$ are obtained from EIS and represents macroscopic measurements, they can reflect the changes in microscopic proton hopping frequency $\nu$ and attempt frequency $\nu_0$, respectively[49,60,61].

If considering proton conduction with and without MIR irradiation at a fixed temperature $T$ as two different processes following the Arrhenius relation, $\Delta\sigma/\sigma$ can be expressed by combining Eq. (2) and (3):

$$\frac{\Delta\sigma}{\sigma} = \left[\frac{\sigma_0^{on}}{\sigma_0^{off}}\exp\left(-\frac{\Delta E_a}{k_BT}\right) - 1\right] \times 100\% \propto \left[\frac{Q^{on}}{Q^{off}}\exp\left(-\frac{\Delta E_a}{k_BT}\right) - 1\right] \times 100\%$$

$$(4)$$

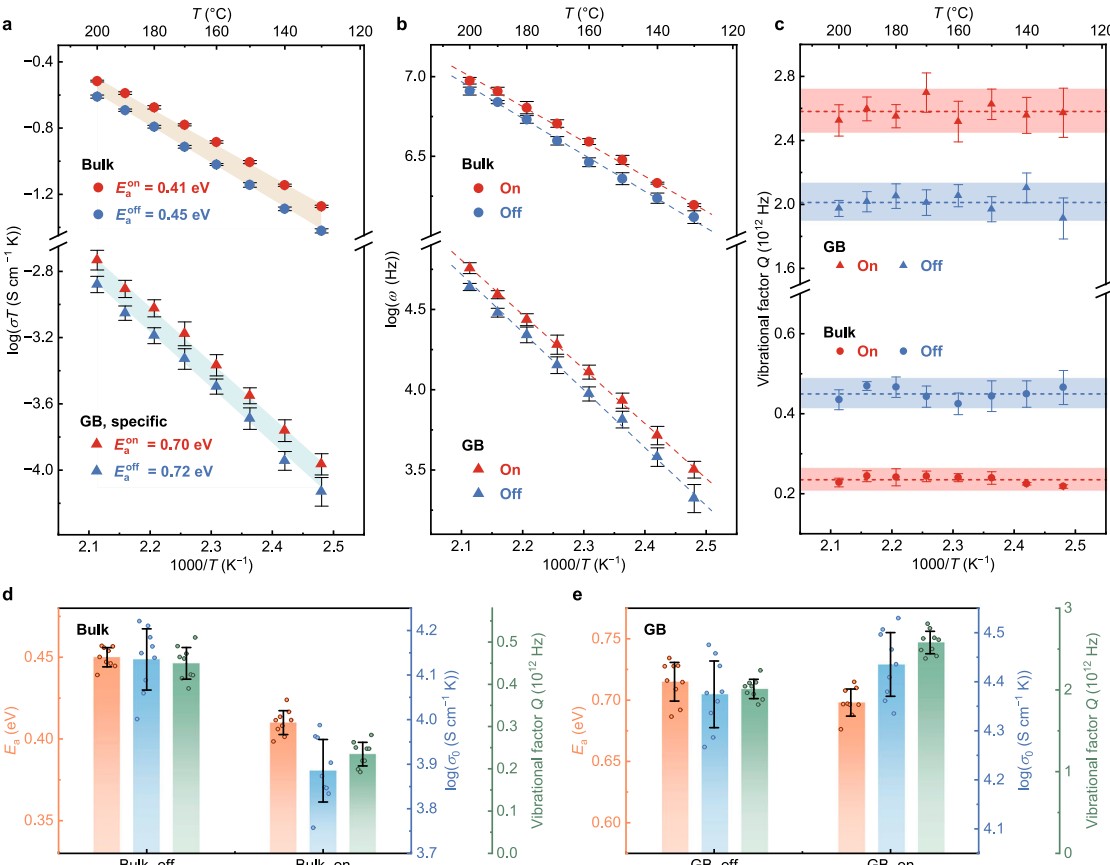

**Fig. 4 | Temperature dependence of the proton conductivity and corresponding parameters with and without MIR irradiation.** Arrhenius plots of (**a**) proton conductivity ($\sigma T$) and (**b**) characteristic frequency of the bulk and GB processes ($\omega$). **c** Vibrational factor ($Q = \omega_0 \exp(\Delta S/k_B)$) of bulk (circle) and GB (triangle) measured with (red) and without (blue) MIR irradiation at 130–200 °C. The calculated $Q$ and their errors are represented by dashed lines and shadowed areas. Comparison of

the corresponding $E_a$ (orange), $\sigma_0$ (blue), and $Q$ (green) of **d** bulk and (**e**) GB. Data in (**a**–**e**) are presented as mean values from three parallel measurements per sample across three samples prepared using the same method. The error bars in (**a**–**e**) incorporate the corresponding standard deviations and fitting errors. Source data are provided in the Source Data file.

where $\sigma_0^{on}/\sigma_0^{off}$ denote the ratios of the prefactors and vibrational factors, and $\Delta E_a = E_a^{on} - E_a^{off}$ refers to the difference in the activation energies with and without MIR irradiation. These parameters indicate the changes in proton conduction properties under different irradiation states. According to Eq. (4), larger $\Delta\sigma/\sigma$ at a given temperature can be achieved by decreasing the activation energy ($\Delta E_a < 0$) or increasing the prefactor ($\sigma_0^{on}/\sigma_0^{off} > 1$). $\Delta\sigma/\sigma$ for the bulk gradually decreases with increasing $T$, when the average thermal energy $k_B T$ contributes more to activate proton hopping. However, in this work, we emphasize that the enhancement effect is larger at low temperatures, where $k_B T$ is much lower than the activation energy for proton hopping, and excitations in atomic vibrations contributing to proton hopping would cause a significant enhancement in proton conductivity.

Figure 4a presents the Arrhenius plot of the bulk and GB proton conductivities with and without MIR irradiation at 130–200 °C. The corresponding $\omega$ and $Q$ are plotted in Fig. 4b, c, respectively. $Q$ were calculated at different temperatures as presented in Fig. 4c for further analysis. The comparison for the trends in $E_a$, $\sigma_0$, and resulting $Q$ are presented in Fig. 4d, e. Quantitative comparison results based on Eq. (4) are summarized in Supplementary Table 5. Upon MIR irradiation, the $E_a$ for bulk proton conduction ($E_{a,Bulk}$) decreased from $0.450 \pm 0.006$ eV to $0.409 \pm 0.007$ eV, with a statistically significant difference of -0.04 eV. Notably, the change in $E_{a,Bulk}$ for the deuterated BZY is 0.02 eV (Supplementary Fig. 7b). The different enhancement ratio (Supplementary Table 3) and activation barrier originate from the greater mass of deuteron and lower energy of O–D stretch vibration.

Conversely, the $E_a$ for GB proton conduction ($E_{a,GB}$) remains unchanged within the error ($0.716 \pm 0.016$ eV and $0.698 \pm 0.011$ eV with and without MIR irradiation, respectively). The $E_a$ data measured without MIR irradiation agree with previous literature[20,49,51]. Despite a reduction in $E_{a,Bulk}$ upon MIR irradiation, the photon energy and optical power density of the MIR radiation are much lower than those required to induce direct flow of protons or lattice atoms[62], again indicating that the absorbed energy of MIR radiation excites the O–H stretch vibration. Notably, both $\sigma_0$ and $Q$ exhibit opposite trends for bulk and GB upon MIR irradiation—for bulk proton conduction, the values decrease ($\sigma_{0,Bulk}^{on} < \sigma_{0,Bulk}^{off}$ and $Q_{Bulk}^{on} < Q_{Bulk}^{off}$); but for GB proton conduction, the values increase ($\sigma_{0,GB}^{on} > \sigma_{0,GB}^{off}$ and $Q_{GB}^{on} > Q_{GB}^{off}$) (Fig. 4d, e). $\sigma_0^{on}/\sigma_0^{on}$ and $Q^{on}/Q^{off}$ are roughly consistent for both bulk and GB (Supplementary Table 5). It can be concluded that the enhancement in bulk proton conductivity of BZY is mainly attributed to the reduction in $E_a$, accompanied by a decrease in $\sigma_0$ and $Q$. On the other hand, the increase in $\sigma_0$ and $Q$ is the major reason for the enhancement in GB proton conductivity.

We further discuss the origin of the distinct behaviours of $\sigma_0$ and $Q$ for bulk and GB proton conductivities, by considering the possible effects of MIR irradiation on the trends for $\nu_0$ and $\Delta S$. For bulk proton conduction, if we take a constant $\omega_0$ (since $\nu_0$ is usually taken as a constant)[63,64] the entropic term $\exp(\Delta S/k_B)$ changes similarly as $\sigma_0$ and $Q$, corresponding to a decrease of $\Delta S$ upon MIR irradiation. The decrease in both $E_a$ and $\Delta S$ obeys the Meyer-Neldel rule, which states that the decrease in $E_a$ is compensated by a contribution to decrease $\Delta S$

for bulk proton conduction[65,66]. Such contribution to $\Delta S$ is associated with the excitations providing the energy for protons to overcome $E_a$, underpinning the role of exciting the O–H stretch vibration in enhancing bulk proton conductivity. In fact, $v_0$ may vary upon the perturbation of the local chemical environment of protons[64], and it is necessary to consider the variation of $v_0$ with and without MIR irradiation. The decrease in $E_a$, $\sigma_0$, and $Q$ has been reported as a consequence of increased lattice anharmonicity and lattice softening, leading to a decrease in $v_0$[64]. Lattice softening also causes a decrease in $\Delta S$. The increased lattice anharmonicity and lattice softening also indicate the correlation between atomic vibrations and the enhancement in proton conductivity. An origin of these effects has been proposed to be the excitation of atomic vibrations altering O–O separation, such as O–H wag and lattice vibrations, thus changing the shape of the potential energy surface (PES)[14,24,64]. It has also been demonstrated that variation of PES caused by different vibration amplitudes accounts for changes in $E_a$ values, where a large amplitude is accompanied by flat PES to facilitate solid-state ionic conduction[25]. Given this, the excitation of atomic vibrations that can modulate the PES of protons can assist proton hopping. Furthermore, exciting O–H stretch vibration with infrared radiation has been shown to influence such atomic vibrations in some proton-conducting oxides, through the coupling of O–H stretch with low-frequency atomic vibrations, as observed by Spahr et al.[35,36] and Sakurai et al.[67]. On the other hand, for GB proton conduction, Fig. 4e shows a slightly increased $\sigma_0$ upon MIR irradiation, indicating that $\Delta S$ increases regardless of constant or decreased $v_0$. This trend can be attributed to the significant configurational entropy on the highly distorted GB region[65]. The change in configurational entropy during proton conduction may be amplified by the excitation of lattice vibrations coupled with O–H stretch vibration, thus could explain the increase in $\Delta S$ for GB proton conduction.

While the aforementioned mechanisms may increase proton mobility, higher proton concentration represents another potential origin of the enhanced proton conductivity. Computational evidence has demonstrated that phonons can influence hydration entropy of acceptor-doped $BaZrO_3$[68,69], thus changing the proton concentration. However, these investigations focused specifically on lattice phonons rather than the O–H stretching vibration central to this work. Furthermore, as shown in Supplementary Fig. 5, the variation in conductivity of the protonated sample was not significant with changing water partial pressure at 160 °C, indicating kinetically limited hydration/dehydration processes at this temperature. On the contrary, the observed MIR-induced conductivity enhancement occurs instantaneously below 200 °C. Given this rapid response, we believe that our observation is not dominated by the light-induced effects on hydration enthalpy and the subsequently increased proton concentration.

Overall, the enhancement in proton conductivity by MIR irradiation may originate from the excitation of O–H stretch vibration. The mechanisms can be that the O–H vibration is excited to a higher energy state with a lower proton hopping barrier and larger vibration amplitude. Such excitation may further reinforce atomic vibrations related to oxygen ions, thus assisting proton hopping. In the following section, we will elaborate on the O–H vibration excitation mechanism.

### Wavelength effect and suggested mechanism

To confirm that the enhanced proton conductivity originates from the excitation of O–H stretch vibration, we further analyse the wavelength effect of MIR radiation. Narrowband MIR radiations of two specific wavelengths are obtained by two different bandpass filters (25 mm diameter, 1 mm thick). As shown in Fig. 5a, the passing wavelength range of filter A is $2.95 \pm 0.055$ μm, which is close to the resonant frequency of O–H stretch vibration. For filter B, the passing wavelength range is $2.70 \pm 0.060$ μm, where only slight absorption was observed. The filtered MIR radiation power incident to the sample was maximized by fixing the filters between the MIR light source and the

sample stage, since the radiation power through the filters is small and decreases with increasing $d$ (Supplementary Fig. 10b). To identify the heating effect of contact heat transfer between the light source, filter, and sample stage, a control group was set up by replacing the filters with a stainless-steel plate (25 mm × 25 mm size, 1 mm thick). $\Delta\sigma/\sigma$ for bulk and GB proton conductivities versus filters A and B and the control group are compared in Fig. 5b. The enhancement effect for filter B is insignificant compared to that of the control group; whereas for filter A, with passing wavelength matching with O–H stretch vibration, the enhancement effect is ~5 times and 3 times in bulk and GB conductivities of the control group, respectively. Such measured MIR light resonant enhancement effect is even greater compared to the estimated results in Fig. 3 (about 2 times of IR heating effect), since the IR heating effect is weaker with narrowband bandpass filtering. We hypothesize that the O–H stretch vibration absorbs the photons matching its frequency in MIR radiation. The resonant photon energy excites the vibrational energy to a higher vibrational state (Fig. 5c and Supplementary Fig. 20a), and promotes proton transfer in the lattice.

We model the effective PES of the proton for a detailed discussion of the above hypothesis. Several studies have elaborated that a proton incorporated into metal oxides vibrates locally around the proton donor ($O_I$ in Fig. 5c–e)[14,36,64]. The photon energy derived from the absorption frequency in the vibrational spectra represents the transition energy of the vibration, normally from the ground state to its first excited state. Following previous works[29,36,67], we model the motion of the proton in the vicinity of $O_I$ by a Morse potential, a typical anharmonic potential (Supplementary Fig. 20a):

$$V(r) = V_0 \left[ 1 - e^{-\alpha(r - r_0)^2} \right] \tag{5}$$

where $r$ is the distance between $O_I$ and H atoms, $r_0$ is the equilibrium $O_I$–H distance, $V_0$ is the activation barrier, $\alpha$ defines the curvature of the potential and can be solved from $v_e$ elaborated below. The energy of the $i$-th vibrational level in the Morse potential is given by $E_i = (i + 1/2) hv_e [1 - \chi (i + 1/2)]$, leading to the transition energy between the $i$-th and $i+1$-th levels $E_{i,i+1} = hv_{i,i+1} = hv_e [1 - 2\chi (i+1)]$. $v_e$ is a theoretical quantity related to the measured frequency of O–H stretch vibration $v_{0,1}$ via $v_e = 1/(1 - 2\chi)$, and $\chi$ is the anharmonic coefficient given by $\chi = hv_e/4V_0$[29,70]. Here, $V_0 = E_0 + E_{a,Bulk}^{eff}$ means the depth of the potential, where $E_0 = 1/2 hv_e (1 - 1/2\chi)$ refers to the ground-state energy, and $E_{a,Bulk}^{eff}$ is the effective activation energy defined as the difference between the top of the activation barrier and $E_0$.

To find $E_i$, we take $E_{a,Bulk}^{eff} = 0.45$ eV (the measured activation energy for bulk proton conduction). This value reasonably reflects the activation barrier for proton conduction of most protons at 130–200 °C, since the majority of protons are reported to remain in a trapped state with an apparent activation barrier of 0.44–0.47 eV at low temperatures[49]. By applying $v_{0,1} = 1 \times 10^{14}$ Hz (O–H stretch frequency from the vibrational spectra) and $r_0 = 1.003$ Å (estimated by an empirical correlation between $v_{O–H}$ and O–H bond length)[71], we obtain $\chi = 0.249$. Therefore, the O–H stretch vibration behaves as an anharmonic oscillator, and the energy gap between its ground state ($E_0$) and the first excited state ($E_1$) is $E_{0,1} = 0.41$ eV. By absorbing an MIR photon matching its vibration frequency, the oscillator is brought to its excited state[70]. This process is analogous to the photoexcitation of electrons or holes in semiconductors[44]. High excitation rate can be achieved by increasing $p$, which promotes the enhancement in proton conductivity. Subsequently, the effective barrier for proton hopping would reduce from 0.45 eV at the ground state to 0.04 eV at the first excited state. comparable with the average thermal energy $k_BT$ in 130–200 °C (0.03–0.04 eV). However, if proton transfer occurs via direct breaking of the O–H bond when the proton is at the first excited state, it would lead to a significant decrease in the activation energy, deviating from our observation that the change in $E_{a,Bulk}$ upon MIR irradiation is only 0.04 eV. Therefore, we propose that the enhanced

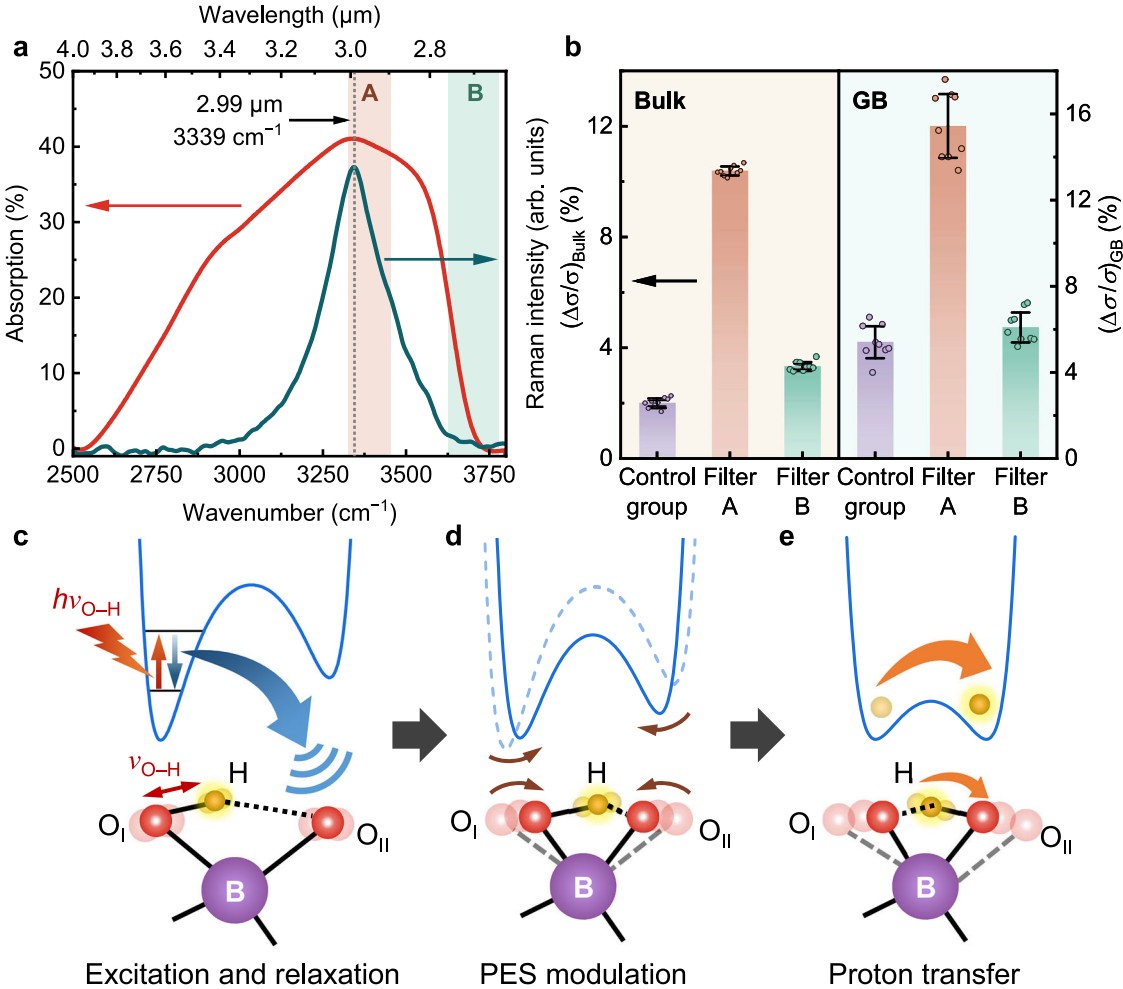

**Fig. 5 | Wavelength effect and suggested mechanism of MIR light enhanced proton conductivity. a** Representation of passing wavelength ranges of the MIR narrowband bandpass filters A (brown shadow) and B (green shadow). The range for filter A is closest to the resonant frequency of O–H stretch vibration. **b** Wavelength effect of MIR light enhanced $\Delta\sigma/\sigma$ of bulk and GB for filters A and B, and the control group. Schematic representations for the lattice vibration-assisted proton hopping process showing (**c**) excitation and relaxation of the O–H stretch vibration, (**d**) modulation of the potential energy surface (PES) of the proton due to subsequent excitation of the coupled lattice vibrations, and (**e**) proton transfer from the donor ($O_I$) to the acceptor ($O_{II}$). For reference, the dashed curve in (**d**) depicts the shape of the PES prior to resonant excitation, corresponding to configuration shown in **c**. The error bars in (**b**) incorporate standard deviations from three parallel measurements and fitting errors per sample across three samples prepared using the same method. Source data are provided in the Source Data file.

proton conduction originates from relaxation of excited O–H stretch vibration into lower-frequency oxygen lattice vibrations coupling to O–H stretch vibration (Fig. 5c). These coupled vibrations alter the O–O distance, thereby tuning the height and width of the PES of the proton (Fig. 5d), and facilitating proton transfer (Fig. 5e)[14,72,73]. For proton-conducting metal oxides, experimental evidence has confirmed that the temperature-dependent excited-state lifetimes of O–H stretch mode of $KTaO_3$ is coupled to the O–Ta–O bending motion, thus reducing the activation barrier height[35,67,74]. Our previous works have also shown that the temperature-dependent proton jump times of BZY and Y-doped $BaCeO_3$ measured by quasielastic neutron scattering follow Samgin's proton polaron model[75–77], suggesting that proton hopping is strongly coupled to lattice phonons, in particular B–O stretch modes[28,46]. The above evidence indicates that the lattice vibrations coupled to O–H stretch vibration are able to tune the PES of the proton, and, in turn, assist proton transfer.

It should be noted that the potential model derived here only depicts the time-averaged effective PES of the proton, since the actual PES shape is dependent on time, position, and temperature due to lattice vibrations and various oxygen chemical environments[14,24]. On the other side, the activation energy for GB proton conduction has a

strong contribution from the high Schottky barrier (up to 0.6 V) due to the depletion of protons in the space-charge region[44,51]. As presented in Supplementary Note 6, upon MIR irradiation, the Schottky barrier height decreases from $0.287 \pm 0.011$ eV to $0.275 \pm 0.009$ eV, resulting in the higher GB enhancement ratio. Although the enhancement effect presented in this work is only up to about 50%, it is worth noting that the transmitted MIR light intensity of the sample (Supplementary Fig. 17b) decreases exponentially against the sample thickness. This thickness dependence suggests that higher enhancement ratios could be achieved in thinner samples.

In conclusion, we enhanced the proton conductivity in protonated polycrystalline BZY proton conductor using CW MIR irradiation with dominant wavelengths of 2–6 μm. Vibrational spectroscopy showed that the radiation between 2.7–4.0 μm can be absorbed by the O–H stretch vibration in BZY. The enhancement in bulk and GB proton conductivities at 160 °C were 36.8% and 53.0% under an effective irradiation power density of 195.2 mW cm⁻². The enhancement effect was reversible when the MIR irradiation was switched on and off for multiple cycles. The results implicate the possibility of an over-ten-fold enhancement of both bulk and grain boundary proton conductivities to over $10^{-3}$ S cm⁻¹ if irradiated with a high-power CW MIR laser.

Conductivity measurements under narrowband MIR irradiation demonstrated that the MIR radiation strongly absorbed by the O–H stretch vibration (2.95 μm) exhibits stronger enhancement in conductivity than that with weak absorption (2.70 μm). The evidence indicates that MIR radiation enhances proton conductivity in BZY by exciting the O–H stretch vibration, which subsequently relax into coupled lattice vibrations that modulate the PES of the proton to lower the energy barrier for proton transfer.

Our findings demonstrate MIR irradiation as a novel strategy to enhance ionic conductivity in fast ionic conductors in addition to existing approaches. This effect holds particular promise for tubular[78,79] and thin-film[80,81] protonic ceramic electrochemical cells (PCEC), with the thickness of electrolyte and air electrode within 400 μm, using IR furnaces[34,56]. This strategy offers advantages including rapid start-up, reduced operation temperature and thermal expansion, power consumption, and costs. Notably, the potential to enhance the ionic conductivity with a low-power light source would inspire the exploration towards low- or even room-temperature PCECs. Furthermore, our findings go beyond conventional chemical approaches to improve the ionic conductivity, and will inspire further exploration of tuning ionic conductivity as well as chemical reaction rates based on selective modulation of vibrational properties of solids.

## Methods

### Sample preparation

The BZY powder was prepared using the conventional solid-state reaction method. The $BaCO_3$ (Macklin, 99.95%), $ZrO_2$ (Adamas-beta, 99%+), and $Y_2O_3$ (Adamas-beta, 99.99%) were mixed in stoichiometric ratios and ball milled in 30 mL isopropanol (Aladdin, AR) for 12 h at 300 rpm, followed by calcination at 1400 °C for 10 h with a ramp rate of 5 °C min$^{-1}$. The resulting product was ball milled again in isopropanol and was sifted with a 200 mesh screen to avoid powder agglomeration and cracking after sintering.

The as-synthesized powder was pressed into pellets under 200 MPa for 30 s. The pellets were ~19 mm in diameter and 1.3 mm in thickness. They were thoroughly buried in BZY powder in an alumina crucible, and were sintered at 1720 °C for 24 h at a ramp rate of 2 °C min$^{-1}$ in air. No sintering additives were used. The relative density of the sintered pellets was over 95%. Protonation of the as-sintered pellets was performed in wet nitrogen at 600 °C for 24 h at a ramp rate of 5 °C min$^{-1}$. The partial pressure of water vapour, $p(H_2O)$, in $H_2O$-saturated (wet) $N_2$ was 0.03 atm, achieved by bubbling $N_2$ through deionized water at 25 °C with a flow rate of 40 mL min$^{-1}$. Sample deuteration was performed following the same procedure and parameters, with deionized water replaced by deuterium oxide ($D_2O$) under identical conditions. Dry samples were prepared by dehydrating the pellets in dry $N_2$ atmosphere with $p(H_2O) < 10^{-4}$ atm at 900 °C for 2 h. Supplementary Table 1 shows that the proton concentration per unit cell was $0.085 \pm 0.002$ measured by Karl-Fischer titration (KEM MKC-710S, in dry $N_2$ at 900 °C). Supplementary Fig. 4 presents the Raman intensity depth profile of O–H stretching band on the cross-section of a protonated BZY pellet, suggesting that the proton concentration at the interior of the pellet is similar compared to that at the surface. Simultaneous thermal analysis (Netzsch STA 449 F3 Jupiter, in dry $N_2$ from room temperature to 900 °C, heating rate 10 °C min$^{-1}$) revealed no significant change in the proton concentration within 130–200 °C (Supplementary Fig. 3).

The pellets were smoothed using abrasive paper after protonation. Two thick, strip Ag electrodes of 3 mm in width each were subsequently painted onto the top side of the samples using silver paste (Sinwe 3701). The spacing between the electrodes was 3 mm to allow MIR radiation onto the samples, and the entire area for proton conduction is ~0.09 cm$^2$. Ag-coated Cu wires were employed to lead the current for EIS measurements.

### Characterization of vibrational properties

The vibrational properties of protonated BZY were characterized through DRIFTS (PerkinElmer Spectrum 100) and confocal Raman spectroscopy (Renishaw inVia Qontor). All the data were measured directly on the protonated BZY pellets at room temperature and controlled humidity (<40% RH). DRIFTS measurements were performed on a 1.1 mm thick sample over the wavenumber range 450–4000 cm$^{-1}$ at a resolution of 2 cm$^{-1}$. The absorption percentage $A$ was derived from $A = 1 - R - T$, where $R$ is the measured diffuse reflectance, and $T$ refers to the transmittance of the sample, which was approximated to be 0 given sufficient thickness. Raman spectra were acquired with a 532 nm laser excitation source for 5 accumulations with 10 s exposure time for each accumulation. The laser power was ~10 mW and was focused through a × 50 L objective lens. The scattered light from the sample was collected with the same objective lens and was dispersed through an 1800 l mm$^{-1}$ grating before detection. The spectral range covered 85–4000 cm$^{-1}$. The DRIFTS and Raman spectra were processed with the proprietary software for the instruments.

### Setup of the lab-made test system

A lab-made test system combining temperature control, MIR irradiation and EIS measurement was designed and fabricated for measurements. Schematic illustrations, control schematic diagrams, and specifications for MIR irradiation and temperature control are displayed in Supplementary Fig. 8–11. All measurements were performed in a gas-tight chamber supplied with ambient air, dry $N_2$ ($p(H_2O) < 10^{-4}$ atm), $H_2O$-saturated $N_2$ (wet $N_2$; $p(H_2O) = 0.02$ atm), and $D_2O$-saturated $N_2$ ($p(D_2O) = 0.02$ atm), respectively. Prior to measurements in each atmosphere, the samples were conditioned in the target atmosphere for over 1 h. The sample fixture consists of a ceramic heating plate, an alumina ring, and Teflon tapes. The sample temperature was monitored, stabilized, and recorded by a thermostat consisting of a PID temperature controller, thermocouple, communication unit, and control software. MIR radiation was generated through a CW MIR light source (Infrasolid HIS550R-AA) with a total optical output power of 140 mW. The emission spectrum centres at 3.34 μm, and >90% of the emitted power is concentrated in 2–6 μm. 26.3% of the total optical output power (36.8 mW) covers the O–H stretching band (2.7–4.0 μm). The radiation is concentrated into an ellipse beam through a parabolic mirror inside the light source. The light source was controlled by a lab-made driver board based on a low dropout voltage regulator. Two different IR narrowband bandpass filters (Edmund Optics) were employed to filter MIR radiation at two specific wavelengths. The filters are 25 mm in diameter and 1 mm in thickness. The passing wavelength ranges of filter A and B are $2.95 \pm 0.055$ μm and $2.70 \pm 0.060$ μm, respectively.

The sample fixture and the MIR light source were mounted to an optical platform. Through a translation stage (Thorlabs), the fixture and the light source were aligned coaxially, and the distance between the sample and the light source ($d$) was precisely adjusted. The entire area for proton conduction on the samples (0.09 cm$^2$ between the electrodes) was exposed to MIR irradiation. The MIR irradiation power density within the O–H stretching band and deposited inside the area was defined as the effective MIR power density ($p$), and was estimated from the angular radiation distribution profile of the light source. $p$ was calibrated as a function of $d$, where $p$ drops as $d$ increases (see Supporting Information for detailed derivation). The minimum of $d$ is 0.9 mm, which was restricted by the geometry of the fixture, and corresponds to the maximum $p$ of 195.2 mW cm$^{-2}$.

### Proton conductivity measurements

Proton conductivity was measured by EIS using a frequency response analyser (FRA, Solartron 1260 A) along with a dielectric interface (Solartron 1296) over the frequency range from 10 MHz to 1 Hz with a sinusoidal voltage of 100 mV amplitude with and without MIR radiation. A combination of equivalent circuit model (ECM) and distribution

of relaxation times (DRT) methods was employed to extract the parameters for bulk, grain boundary and electrode polarization processes with reduced fitting errors (Supplementary Note 5)[82,83]. Single-frequency impedance measurements were performed on the FRA with an amplitude of 300 mV and a sampling rate of 1 s (upper limit of the FRA). This technique is also known as single-frequency impedance transients (SFITs)[44]. In the EIS results of this work, $Z'$ (the real part of impedance) at 150 kHz and 1 kHz can reflect bulk and GB features in the spectra, respectively. Therefore, $Z'$ at these two frequencies was successively measured over time while repeatedly switching the MIR light on and off, where the durations of on or off time were both 150 s.

## Supplementary information

Characterization of crystal structure, microstructure, and proton concentration and distribution; effects of atmosphere, proton concentration, and H/D isotopes on proton conductivities; additional information about the lab-made test system; additional EIS results; EIS spectra analysis combining equivalent circuit model (ECM) and distribution of relaxation times (DRT); model for the grain boundary (GB) conductivity; thickness-dependent MIR intensity distribution in the samples; impact of IR heating effect on sample temperature; impact of thermal expansion on sample geometry; effective potential energy surface (PES) of the proton (PDF).

## Reporting summary

Further information on research design is available in the Nature Portfolio Reporting Summary linked to this article.

## Data availability

The data generated in this study are available in the main article, Supplementary Information, and Source Data file. Source data are provided with this paper.

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

## Acknowledgements

The authors acknowledge the financial support from the National Natural Science Foundation of China (52272227 and 12274295) and the Natural Science Foundation of Shanghai (22ZR1428800). H.L. and Q.C. thank Hong Zhu, Shouhang Bo (Shanghai Jiao Tong University), and Qiyang Lu (Westlake University) for their suggestions. The authors thank three anonymous reviewers for providing valuable comments.

## Author contributions

H. L., W. W., and Q. C. conceived the project and designed the experiments. H. L., Y. Z., and R. M. built the lab-made test system. H. L., Y. Z., Z. Z., and J. L. performed the experiments and analysed the experimental data. W. W. and Q. C. provided suggestions on the experiments. H. L., Z. Z., W. W., and Q. C. wrote the manuscript. All authors discussed the results and commented on the manuscript.

## Competing interests

The authors declare no competing interests.
