## [Transparent Peer Review file · Nature Communications]

Mid-infrared light resonance-enhanced proton conductivity in ceramics

Corresponding Author: Dr Qianli Chen

Version 0:

Reviewer comments:

Reviewer #1

(Remarks to the Author)

In this manuscript, Li et al. demonstrated that mid-infrared light can control the proton conductivity in Y-doped BaZrO₃ ceramics by resonant excitation of the O-H stretch vibration. To eliminate the infrared-induced heating effect, the authors built a test system to maintain the temperature constant. By analyzing the activation energies and prefactors, they claimed that the mid-infrared excitation reduces the activation energy and attempts the frequency of proton hopping. A stronger enhancement ratio was found at the O-H stretch frequency, confirming the resonant excitation of O-H vibration. Light-induced ionic conductivity enhancement is an important topic relevant to various types of solid-state ionic conductors. In this work, the infrared light source used has a power much smaller than that of conventional furnaces. The manuscript contains an in-depth analysis, and the results are interesting. The authors should consider addressing the following issues:

1. The enhancement ratio of grain boundaries is higher than that of bulk conductivity, both without and with the bandpass filter. The reason for the higher enhancement in grain boundaries should be clarified.
2. The calculated heating effect in Fig. 3c and d (1/3 to 1/2 of the enhancement ratio) is less than the measured results in Fig. 5b (control and Filter B show about 2/3 enhancement ratio than the resonant excited Filter A). This discrepancy between the calculated and measured results should be explained.
3. One problem with the current PCECs is that the thermal expansion coefficient of the electrode materials does not match that of the electrodes. Can the authors comment on whether this mid-infrared-induced effect causes material expansion?
4. The presented enhancement ratio of about 40% or less is still far for this effect to be practically applied. Moreover, due to the small penetration depth of infrared light in the material, the application of this effect in PCECs can be limited. If the penetration depth cannot be improved, the application of this effect might be limited to thin film devices. But how could it be possible to increase the penetration depth in the devices?

Reviewer #2

(Remarks to the Author)

The work by Li et al. explores the idea of using photons to enhance proton transport in a barium zirconate. I find the idea interesting, and the concept is timely, a lot of groups are thinking about light-driven ionic conduction. I am not against publication, but in the current state I am not convinced that the effect the authors see is fully justified with the data and the analyses. I do have a few comments that I think need to be addressed.

- 1) The authors impedance data clearly shows changes between on- and off-states suggesting indeed a potential increase in conductivity. However, for proton conduction, one would expect to test these experiments under additional experimental conditions to confirm the bulk transport and not changes in the surface (see impedance comment) etc.
 - a. conducting the same measurements at different partial pressures of water,
 - b. conducting the experiments with deuterated samples as deuteration should change the potential well.
- 2) What is the penetration depth of the MIR. The pellets are quite thick, is it clear that we have full photon bulk penetration?
- 3) Impedance:

- a. the authors state that the use of their “three serial R(CPE) networks” indicate proton conduction. No, these EIS are fit with an equivalent circuit assuming proton conduction. Phrasing is important.
- b. The impedance of the GB is quite broad and not well resolved. My concern is that any change in process may have led to inductivity – which would ultimately move the bulk process to lower measured resistances and only suggesting faster transport. What are the capacitances of the process, what are the ideality factors of the bulk process. Have these changed?
- c. How reliable is the change in bulk transport? since the processes are not so well resolved, can the change in GB transport just have affected any bulk response? What does a DRT analysis suggest? The change in GB resistance is much higher than the bulk, ultimately leading to convoluted impedance.
- d. Considering that the electrode process is not well resolved, changes in the electrode would affect the GB process here as well. A surface enhancement of transport has not been ruled out here.
- 4) The authors check if T increase from IR heating would affect it and conclude that the effect is 2-3 times larger. How significant is that and does it neglect thermal expansion of the material?
- 5) Have the authors considered that the light affects the defect formation enthalpy rather than the actual diffusion thermodynamics of the process? Basically, is it the migration enthalpy that is affected or the defect formation enthalpy?
- 6) There seems to be a misconception about the apex frequency in the impedance. On page 10, the authors use the apex frequency of the bulk process ($\omega = 1/RC$) and suggest that this is the jump frequency ν . These are two entirely different parameters
- 7) Figure 4 a gives the bulk and grain boundary conductivities. Grain boundary conductivities cannot be calculated from the impedance data without the in-depth knowledge of the microstructure. See J Power Source 2011, 196, 6456. If these information are not known, one can only plot the resistances but not the conductivity. Arrhenius plots and activation barriers can be generated from the inverse resistance.
- 8) Should the proposed mechanism not lead to a change in activation barrier? But this is experimentally not observed, the proposed mechanism would suggest it.
- 9) What do the authors mean with “The fluctuation of the configurational entropy may be amplified”. What is a fluctuation of a configurational entropy in such a disordered proton conductor?

Reviewer #3

(Remarks to the Author)

It is highly meaningful to find and establish some efficient approaches to improve ionic conduction. As stated by the authors, MIR-induced enhancement of proton conduction has been reported but scarce (refs 41 and 42). This work demonstrates experimentally that this method is quite effective for proton conductor. I think this article is suitable for publication in Nat Commun after careful clarification.

1. The test was conducted in ambient air. As reported by many reports, the air humidity has a large impact on proton conductivities. This impact may be more remarkable when the MIR light is on. A valuation in N₂ or vacuum is suggested.
2. I am very curious about the homogeneity of protons in the material and its effect on MIR-induced enhancement of the proton conductivities. The protonated BZY was prepared as “Protonation of the as-sintered pellets was performed in wet nitrogen at 600 °C for 24 h at a ramp rate of 5 °C min⁻¹”. It is possible that the concentration of the proton atom decreases from the surface to the interior. As shown in the test setup (Fig. 2a), the two Ag electrodes were placed directly on the upper surface of the pellet. The conduction path on the surface would be dominant supposing the higher H concentration on the surface.
3. The authors tried to exclude the impact of MIR-induced thermal effect. The monitoring of the pellet temperature was on the lower surface of the material. Likewise, as shown in Fig. 2a, the MIR light was directly shined to the upper side of the material and the two electrodes. In this case, the thermal diffusion from the upper side to the bottom may be not so rapid as we imagine. So, the thermal effect should be carefully re-valuation. Test the temperature of the upper surface?
4. The symbol “GB” should be noted when its full name first appears.

I am not an expert on the theoretical aspect described in the manuscript, which requires valuation by other referees.

Version 1:

Reviewer comments:

Reviewer #1

(Remarks to the Author)

I am satisfied with the modifications that the authors made. I think the paper can be accepted in the current form.

Reviewer #2

(Remarks to the Author)

The authors have adequately responded to my comments and performed a significant number of additional experiments and analyses to corroborate their finding. I can recommend publication.

Reviewer #3

(Remarks to the Author)

The article has been revised well according to all comments, and publication is suggested.

Response to reviewers' comments

We are submitting the revised version of our manuscript entitled “Mid-infrared light resonance-enhanced proton conductivity in ceramics” (Manuscript ID: NCOMMS-24-65739). We are grateful to the reviewers who have taken their time to read our manuscript carefully, with scientific interest and provide valuable comments. In the revised manuscript, to address the reviewers' comments, **we performed the extra experiments on thinner pellets and in various atmospheres to confirm our observation. We also performed DRT analysis to better resolve the contribution of grain boundary (GB) response to the measured impedance spectra, and calculated the specific grain boundary conductivity that considers the material microstructure.** Please find below a detailed point-by-point response to the reviewers' comments. Following the “Author reply” below, we listed the most important changes made in light of the reviewers' comments. Complete modifications have been highlighted in the marked-up version of our manuscript and Supplementary Information. In addition, we have read through the manuscript carefully, made changes for clarification, and corrected for grammar and typos.

Black text = reviewers' comments

Blue text = our response (**bold text = major changes or action taken in this revision**)

Highlighted text = inserted into the revised manuscript

Text highlighted in green = inserted into the revised Supplementary Information

Reviewer #1

Comments of Reviewer #1:

In this manuscript, Li et al. demonstrated that mid-infrared light can control the proton conductivity in Y-doped BaZrO₃ ceramics by resonant excitation of the O-H stretch vibration. To eliminate the infrared-induced heating effect, the authors built a test system to maintain the temperature constant. By analyzing the activation energies and prefactors, they claimed that the mid-infrared excitation reduces the activation energy and attempts the frequency of proton hopping. A stronger enhancement ratio was found at the O-H stretch frequency, confirming the resonant excitation of O-H vibration. Light-induced ionic conductivity enhancement is an important topic relevant to various types of solid-state ionic conductors. In

this work, the infrared light source used has a power much smaller than that of conventional furnaces. The manuscript contains an in-depth analysis, and the results are interesting.

Author reply: Thank you for your encouraging remarks and the valuable comments. Please see our response to your detailed comments below:

Comment 1 of Reviewer #1:

The enhancement ratio of grain boundaries is higher than that of bulk conductivity, both without and with the bandpass filter. The reason for the higher enhancement in grain boundaries should be clarified.

Author reply: We consider that the high enhancement ratio of grain boundary conductivity originates from the **lower Schottky barrier height in the grain boundary space charge region upon MIR irradiation, as analysed in detail below.**

Changes in the manuscript:

Page 19:

As presented in Supplementary Note 6, upon MIR irradiation, the Schottky barrier height decreases from 0.287 ± 0.011 eV to 0.275 ± 0.009 eV, resulting in the higher GB enhancement ratio.

Changes in the Supplementary Information:

Page S17:

Here, σ_{GB} is approximately two orders of magnitude lower than the bulk conductivity (σ_{Bulk}), primarily due to proton depletion in the space charge layer.^{22,23} The GB structure comprises a positively charged core – enriched with oxygen vacancies and protons to accommodate misfit strain – flanked by two negatively charged space charge layers that electrostatically balance the positive charge of GB core.²³ Through the core region, Yang et al.²⁴ and Bondevik et al.²⁵ indicated that strong structural distortion and proton segregation at the GB core significantly lower proton mobility. According to the space charge model, the subsequent potential difference between the bulk and the GB core is denoted by ϕ_B , also known as Schottky barrier height. Kjølseth et al.²³ and Jiang et al.²⁶ have shown that, when expressed in terms of ϕ_B , σ_{GB} follows the relation

$$\begin{aligned}\sigma_{\text{GB}} &= \sigma_{\text{Bulk}} \frac{2e\varphi_{\text{B}}}{k_{\text{B}}T} \exp\left(\frac{-e\varphi_{\text{B}}}{k_{\text{B}}T}\right) \\ &= \sigma_{0,\text{Bulk}} \frac{2e\varphi_{\text{B}}}{k_{\text{B}}T^2} \exp\left[\frac{-(E_{\text{a,Bulk}} + e\varphi_{\text{B}})}{k_{\text{B}}T}\right]\end{aligned}\quad (\text{S3})$$

where σ_{Bulk} , $\sigma_{0,\text{Bulk}}$ and $E_{\text{a,Bulk}}$ represent bulk proton conductivity, and the corresponding prefactor and activation energy. φ_{B} can be determined by subtracting the $E_{\text{a,Bulk}}$ value from $(E_{\text{a,Bulk}} + \varphi_{\text{B}})$ by fitting $\ln(\sigma_{\text{GB}} T^2)$ versus $1000 T^{-1}$.

As indicated by Eq. (1) in the main text, the on-off ratio of proton conductivity ($\sigma^{\text{on}}/\sigma^{\text{off}}$) governs the enhancement ratio ($\Delta\sigma/\sigma$). For GB proton conduction, this ratio can be explicitly expressed using Eq. (S3) for a fixed temperature T :

$$\begin{aligned}\frac{\sigma_{\text{GB}}^{\text{on}}}{\sigma_{\text{GB}}^{\text{off}}} &= \frac{\sigma_{0,\text{Bulk}}^{\text{on}}}{\sigma_{0,\text{Bulk}}^{\text{off}}} \frac{\varphi_{\text{B}}^{\text{on}}}{\varphi_{\text{B}}^{\text{off}}} \exp\left[\frac{-(\Delta E_{\text{a,Bulk}} + e\Delta\varphi_{\text{B}})}{k_{\text{B}}T}\right] \\ &= \frac{\sigma_{\text{Bulk}}^{\text{on}}}{\sigma_{\text{Bulk}}^{\text{off}}} \frac{\varphi_{\text{B}}^{\text{on}}}{\varphi_{\text{B}}^{\text{off}}} \exp\left(-\frac{e\Delta\varphi_{\text{B}}}{k_{\text{B}}T}\right)\end{aligned}\quad (\text{S4})$$

where $\Delta E_{\text{a,Bulk}} = E_{\text{a,Bulk}}^{\text{on}} - E_{\text{a,Bulk}}^{\text{off}}$ and $\Delta\varphi_{\text{B}} = \varphi_{\text{B}}^{\text{on}} - \varphi_{\text{B}}^{\text{off}}$. When $\varphi_{\text{B}}^{\text{on}} < \varphi_{\text{B}}^{\text{off}}$, both $\varphi_{\text{B}}^{\text{on}}/\varphi_{\text{B}}^{\text{off}} < 1$ and $\Delta\varphi_{\text{B}} < 0$, resulting in a higher on-off ratio and greater enhancement ratio of GB proton conductivity compared to bulk.

For the 0.4-mm thick sample in wet N_2 , φ_{B} showed a subtle but consistent decrease from 0.287 ± 0.011 eV (without MIR irradiation) to 0.275 ± 0.009 eV (with MIR irradiation). The lowered φ_{B} may result from mitigated proton segregation at the GB core, as MIR irradiation could enhance proton mobility. This reduction in Schottky barrier height partially explains the greater enhancement ratio of GB proton conductivity with MIR irradiation. By inserting the calculated $\Delta\varphi_{\text{B}}$ and $\varphi_{\text{B}}^{\text{on}}/\varphi_{\text{B}}^{\text{off}}$ values (derived from the φ_{B} analysis) along with $\sigma_{\text{Bulk}}^{\text{on}}/\sigma_{\text{Bulk}}^{\text{off}}$ (obtained from $(\Delta\sigma/\sigma)_{\text{Bulk}}$ at 160 °C in the main text) into Eq. (S4), we obtain $\sigma_{\text{GB}}^{\text{on}}/\sigma_{\text{GB}}^{\text{off}} = 1.72$. This corresponds to $(\Delta\sigma/\sigma)_{\text{GB}} = 72\%$, which is in acceptable agreement with the experimental value (53%) and confirms the model's consistency.

Comment 2 of Reviewer #1:

The calculated heating effect in Fig. 3c and d (1/3 to 1/2 of the enhancement ratio) is less than the measured results in Fig. 5b (control and Filter B show about 2/3 enhancement ratio than the resonant excited Filter A). This discrepancy between the calculated and measured results should be explained.

Author reply: This difference between the calculated and measured results arises from

different experimental configurations. In Fig. 3, the heating effect was calculated according to the data we measured without narrowband bandpass filtering, while the results in Fig. 5 were measured with the filtering. In the revision, we present the data from a thinner pellet of about 0.4 mm. The revised Fig. 5b shows higher enhancement ratio. The measured heating effect (~20% of the enhancement ratio for bulk) is lower than the calculated results (~40%) in Fig. 3, because more irradiated MIR energy can be converted to heat without bandpass filtering.

Changes in the manuscript:

Page 15-16:

... for filter A, with passing wavelength matching with O–H stretch vibration, the enhancement effect is approximately 5 times and 3 times in bulk and GB conductivities of the control group, respectively. Such measured MIR light resonant enhancement effect is even greater compared to the estimated results in Fig. 3 (about 2 times of IR heating effect), since the IR heating effect is weaker with narrowband bandpass filtering.

Comment 3 of Reviewer #1:

One problem with the current PCECs is that the thermal expansion coefficient of the electrode materials does not match that of the electrodes. Can the authors comment on whether this mid-infrared-induced effect causes material expansion?

Author reply: To the best of our knowledge, there are no studies reporting that optical effects of infrared light can cause material expansion. Therefore, we consider the thermal expansion to be only caused by radiative heating (IR heating). We find that to reach the same proton conductivity, the sample temperature rise due to IR heating (about 5 °C) is lower than the value caused by thermal heating (about 13 °C). We also use the thermal expansion coefficient of BZY to estimate the degree of thermal expansion, according to this IR heating-induced temperature rise. Please see the detailed results in **Supplementary Notes 8-9**.

Changes in the manuscript:

Page 9:

To reach the same proton conductivity, the temperature rise and thermal expansion due to IR heating is still less than the effect caused by thermal heating (Supplementary Notes 8–9).

Changes in the Supplementary Information:

Page S19:

Supplementary Note 8: Impact of IR heating effect on sample temperature

The heating effect of MIR irradiation on the samples is manifested as the percent change in conductivity ($\Delta\sigma/\sigma$) due to the temperature rise (ΔT_h) upon irradiation of light. ΔT_h is defined by the difference between: (i) the steady-state sample temperature during MIR irradiation while deactivating the PID control, and (ii) the reference temperature T . With the PID feedback loop deactivated, the thermostat's output power was fixed at its baseline level (without MIR irradiation). In this configuration, all observed sample temperature variations can be reasonably attributed to IR heating effect. As plotted in Supplementary Fig. 18, ΔT_h was found to be approximately 5 °C when T is 160 °C. The resulting $(\Delta\sigma/\sigma)_{\text{Bulk}}$ and $(\Delta\sigma/\sigma)_{\text{GB}}$ were estimated to be 15% and 25%, respectively, which are lower than those observed in experiment (36.8% and 53.0%). On the other hand, achieving the observed $\Delta\sigma/\sigma$ values solely by heating would require $\Delta T_h = 13$ °C. Such contrast demonstrates a higher energy efficiency of MIR irradiation, achieving a comparable enhancement ratio in proton conductivity at significantly lower temperatures.

To quantify ΔT_h as a function of working distance (d), we modeled the sample as a thermal resistor considering only conductive heat transfer. We further assumed complete conversion of incident optical power to heat flow (P_h), thus having $\Delta T_h \propto P_h$.²² The thickness-dependent P_h was calculated by scaling the normalized function $p(d)$ (Supplementary Fig. 10b) to the range [0,1]. As shown in Figures 3c, d of the main text, the observed MIR-induced enhancement in proton conductivity ($\Delta\sigma/\sigma$) for both bulk and grain boundaries exceed the heating-only estimates by a factor of 2–3. This significant discrepancy demonstrates that IR heating effect cannot account for the majority of the enhancement in proton conductivity.

Supplementary Fig. 18. Representative sample temperature curve upon switching on the MIR light source and deactivated PID feedback loop. The temperature rise (ΔT_h) of the sample at equilibrium is approximately 5 °C when the reference temperature is 160 °C. The resulting $(\Delta\sigma/\sigma)_{\text{Bulk}}$ and $(\Delta\sigma/\sigma)_{\text{GB}}$ were estimated to be 15% and 25%, which are lower than those observed in experiment (36.8% and 53.0%).

Supplementary Note 9: Impact of thermal expansion on sample geometry

In protonic ceramic electrochemical cells (PCECs), thermal expansion coefficient mismatch between electrode and electrolyte materials generates thermal stress under operating conditions. Such stress can initiate microcracks in the cell stack, ultimately leading to significant performance degradation.²⁸ The impact of thermal expansion with and without MIR irradiation was then evaluated.

To the best of our knowledge, there are no studies reporting material expansion due to optical effects of infrared light. Therefore, we attribute the thermal expansion solely to IR heating effects. The thermal expansion coefficient (α) of BZY20 has been reported to be $8.2 \times 10^{-6} \text{ K}^{-1}$.²⁹ For an isotropic material with initial volume V_0 at the reference temperature, the thermally expanded volume due to temperature rise ΔT_h is given by $V_T = V_0 (1 + \alpha\Delta T_h)^3$, yielding the volume ratio $V_T/V_0 = (1 + \alpha\Delta T_h)^3$.²⁶

On the other hand, the proton conductivity σ is expressed as

$$\sigma = \frac{1}{R} \frac{d}{A} = \frac{1}{R} F \quad (\text{S5})$$

where R , d , and A denote the sample's resistance, length and cross-sectional area, and $F = d/A$ is defined as a geometric factor. By analogy, the geometric factor ratio induced by ΔT_h , $F_T/F_0 = 1/(1 + \alpha\Delta T_h)$ can be obtained.

As discussed in Supplementary Section 8, ΔT_h is approximately 5 °C when the reference temperature is 160 °C. The corresponding changes in V_T/V_0 and F_T/F_0 are both less than

0.001 (Supplementary Fig. 19), confirming that thermal expansion effects are negligible under MIR irradiation. Moreover, 5 °C is lower than the ΔT_h required to achieve the observed $\Delta\sigma/\sigma$ values only through heating (13 °C), implying significantly reduced thermal expansion, and hence thermal stress. These findings validate two key conclusions: (i) Eq. (1) in the main text is valid as the change in sample geometry is within 0.1% during MIR irradiation, and (ii) MIR irradiation would achieve significant proton conductivity enhancement while inducing minimal thermal stress in PCECs – a crucial advantage over conventional heating approach.

Supplementary Fig. 19. a Volume ratio (V_T/V_0) and **b** geometric factor ratio (F_T/F_0) for calculating proton conductivity showing marginal changes due to IR heating effect.

Comment 4 of Reviewer #1:

The presented enhancement ratio of about 40% or less is still far for this effect to be practically applied. Moreover, due to the small penetration depth of infrared light in the material, the application of this effect in PCECs can be limited. If the penetration depth cannot be improved, the application of this effect might be limited to thin film devices. But how could it be possible to increase the penetration depth in the devices?

Author reply: In the revision, we measured the relative transmitted intensity of MIR light through the BZY samples using a high-sensitivity MIR detector and our MIR light source (see details in **Supplementary Note 7**). Since this intensity decays exponentially with respect to sample thickness, the penetration of MIR over the thickness of 0.4 mm is still limited. Increasing the penetration depth of MIR light may be difficult. However, the thickness of electrolyte & air electrode of current PCECs commonly reach below 0.4 mm. Therefore, we believe this strategy can be applied to **tubular and thin film PCECs**.

Changes in the manuscript:

Page 19:

This effect holds particular promise for tubular^{78,79} and thin-film^{80,81} protonic ceramic electrochemical cells (PCEC), with the thickness of electrolyte and air electrode within 400 μm , using IR furnaces.^{34,56}

Reviewer #2

Comments of Reviewer #2:

The work by Li et al. explores the idea of using photons to enhance proton transport in a barium zirconate. I find the idea interesting, and the concept is timely, a lot of groups are thinking about light-driven ionic conduction. I am not against publication, but in the current state I am not convinced that the effect the authors see is fully justified with the data and the analyses.

Author reply: Thank you for the careful review and valuable comments. We have made revisions according to your questions and comments. We hope that our response and the revised manuscript can satisfactorily address your concern.

Comment 1 of Reviewer #2:

The authors impedance data clearly shows changes between on- and off-states suggesting indeed a potential increase in conductivity. However, for proton conduction, one would expect to test these experiments under additional experimental conditions to confirm the bulk transport and not changes in the surface (see impedance comment) etc.

- a. conducting the same measurements at different partial pressures of water,
- b. conducting the experiments with deuterated samples as deuteration should change the potential well.

Author reply: Thank you for your valuable suggestions. In the revised manuscript, we performed the experiments again in a gas-tight chamber. **Figs. 2–5** in the manuscript were updated according to results from 0.4-mm-thick pellets in wet N_2 .

- a. We first compared the results of the protonated sample in H_2O -saturated N_2 (wet N_2 ; $p(\text{H}_2\text{O})$)

= 0.02 atm) and dry N₂ ($p(\text{H}_2\text{O}) < 10^{-4}$ atm). The impact of water partial pressures was small for the protonated sample in the temperature range of our experiment (**Supplementary Fig. 10**). The changes in impedance were not significant when changing water partial pressure either with or without MIR irradiation, but the enhancement is prominent when the MIR light was switched on, confirming that the measured enhancement in electrical conductivity originates from bulk proton transport, but not a surface protonic effect.

b. Then, we measured and compared the results in the following conditions: protonated sample at $p(\text{H}_2\text{O}) = 0.02$ atm, deuterated sample at $p(\text{D}_2\text{O}) = 0.02$ atm, and dry sample with $p(\text{H}_2\text{O}) < 10^{-4}$ atm. High proton transport number (**Supplementary Table 2**) and H/D isotope effect (**Supplementary Fig. 9**) confirmed the dominant Grotthuss-type proton conduction in the sample. The enhancement ratios of bulk and specific GB conductivities for deuterated and dry samples were smaller than those of the protonated sample (**Supplementary Table 4**), indicating the observed effect originates from enhanced proton conductivity.

Changes in the manuscript:

Page 7:

Additionally, no significant changes were observed in the EIS spectra measured in wet N₂, ambient air, and dry N₂ atmospheres (Supplementary Fig. 5), indicating a negligible atmosphere effect at 160 °C. The changes in impedance were not significant when changing water partial pressure either with or without MIR irradiation, suggesting that the conductivity is not a surface protonic effect.⁵⁴ Nevertheless, the enhancement was prominent when the MIR light was switched on, confirming that the measured enhancement in electrical conductivity originates from bulk transport.

Page 10:

To confirm that the observed effect originates from enhanced proton conductivity, we further conducted the experiments on deuterated samples in D₂O-saturated N₂ ($p(\text{D}_2\text{O}) = 0.02$ atm) and dry samples in dry N₂ ($p(\text{H}_2\text{O}) < 10^{-4}$ atm). As shown in Supplementary Table 2, the enhancement ratios of both bulk and GB conductivities for the deuterated samples are 23.1% and 33.6%, approximately 0.6 times those of the protonated samples due to the heavier deuteron. For the dry samples, the enhancement ratios are only 3.91% and 6.54%. These results confirm that the enhancement in conductivity is dominated by contributions specific to protons.

Page 12:

Notably, the change in $E_{a,Bulk}$ for the deuterated BZY is 0.02 eV (Supplementary Fig. 7b). The different enhancement ratio (Supplementary Table 3) and activation barrier originate from the greater mass of deuteron and lower energy of O–D stretch vibration.

Page 20:

Protonation of the as-sintered pellets was performed in wet nitrogen at 600 °C for 24 h at a ramp rate of 5 °C min⁻¹. The partial pressure of water vapor, $p(\text{H}_2\text{O})$, in H₂O-saturated (wet) N₂ was 0.03 atm, achieved by bubbling N₂ through deionized water at 25 °C with a flow rate of 40 mL min⁻¹. Sample deuteration was performed following the same procedure and parameters, with deionized water replaced by deuterium oxide (D₂O) under identical conditions. Dry samples were prepared by dehydrating the pellets in dry N₂ atmosphere with $p(\text{H}_2\text{O}) < 10^{-4}$ atm at 900 °C for 2 h.

Page 21:

All measurements were performed in a gas-tight chamber supplied with ambient air, dry N₂ ($p(\text{H}_2\text{O}) < 10^{-4}$ atm), H₂O-saturated N₂ (wet N₂; $p(\text{H}_2\text{O}) = 0.02$ atm), and D₂O-saturated N₂ ($p(\text{D}_2\text{O}) = 0.02$ atm), respectively. Prior to measurements in each atmosphere, the samples were conditioned in the target atmosphere for over 1 h.

Changes in the Supplementary Information:

Page S6:

Supplementary Note 2: Effects of atmosphere, proton concentration, and H/D isotopes on proton conductivities

Supplementary Fig. 5. Nyquist plots of protonated BZY samples measured at 160 °C in

H₂O-saturated (wet) N₂, ambient air, and dry N₂ with MIR irradiation **a** off and **b** on. The insets illustrate the magnification of bulk features. The atmosphere in the gas-tight chamber was switched from H₂O-saturated N₂ to ambient air, then to dry N₂. The target atmosphere was purged into the chamber for over 1 h at a high gas flow rate before impedance measurements were taken. No significant changes were observed in the impedance spectra in different atmospheres, suggesting no significant hydration/dehydration processes at 160 °C.

Supplementary Table 2. Bulk proton conductivity without MIR irradiation in protonated ($\sigma_{\text{Bulk,wet}}$) and dry ($\sigma_{\text{Bulk,dry}}$) samples, and proton transport number (t_{H}) at 160 °C. The protonated and dry samples were prepared separately, as described in Methods. The proton transport number was calculated by $t_{\text{H}} = 1 - \sigma_{\text{Bulk,dry}}/\sigma_{\text{Bulk,wet}}$.⁵ The calculated t_{H} , close to unity, confirms the dominant proton conduction in BZY20.

Condition	Atmosphere	σ_{Bulk} (S cm ⁻¹)	t_{H}
Protonated	Wet N ₂	$(2.20 \pm 0.03) \times 10^{-4}$	0.975 ± 0.001
Dry	Dry N ₂	$(5.53 \pm 0.11) \times 10^{-6}$	

Supplementary Fig. 6. Raman spectra of O–H and O–D stretch vibration bands ($\nu_{\text{O-H}} = 3333 \text{ cm}^{-1}$; $\nu_{\text{O-D}} = 2472 \text{ cm}^{-1}$) in protonated and deuterated BZY20 samples. The observed results agree with literature.⁶ The isotope frequency shift ($\nu_{\text{O-H}}/\nu_{\text{O-D}}$) of 1.347, lower than 1.374 predicted by the harmonic potential model, suggests significant anharmonicity in the potential governing proton transfer.^{7,8}

Supplementary Fig. 7. Arrhenius plots of **a** bulk conductivity (σT) of protonated (H, circles) and deuterated (D, open circles) samples measured without MIR irradiation, and **b** deuterated sample measured with (red) and without (blue) MIR irradiation at 130–200 °C. The difference in the activation energies and the ratio of the prefactors of protonated and deuterated samples are marked in **a**, indicating a clear H/D isotope effect.⁶

Supplementary Table 3. Enhancement ratio of protonated samples at $p(\text{H}_2\text{O}) = 0.02$ atm, deuterated samples at $p(\text{D}_2\text{O}) = 0.02$ atm, and dry samples with $p(\text{H}_2\text{O}) < 10^{-4}$ atm.

Condition	$(\Delta\sigma/\sigma)_{\text{Bulk}}$ (%)	$(\Delta\sigma/\sigma)_{\text{GB}}$ (%)
Protonated	36.8±0.51	53.0±6.75
Deuterated	23.1±0.61	33.6±12.2
Dry	3.91±0.14	6.54±0.92

Page S8:

Supplementary Note 3: Additional information about the lab-made test system

Supplementary Fig. 8. Schematic illustration and optical image for setups **a** without and **b** with the IR bandpass filter of the lab-made test system. d denotes working distance (the distance between top surface of the sample and emission window of the light source). The bandpass filter is in contact with sample fixture and the light source. **c** Optical image of the gas-tight chamber enclosing the lab-made test system with internal dimensions of 15 cm × 10 cm × 20 cm. Different atmospheres were obtained by purging the chamber with the target atmospheres for over 1 h.

Supplementary Fig. 9. Schematic diagram of the lab-made test system, and optical image of the BZY sample. Black arrows indicate the direction of electrical signals. The feedback loop between the heating plate, sample and temperature controller stabilizes sample temperature.

Comment 2 of Reviewer #2:

What is the penetration depth of the MIR. The pellets are quite thick, is it clear that we have full photon bulk penetration?

Author reply: Following your question, we measured the relative transmitted intensity of MIR light source through the BZY samples using a high-sensitivity MIR detector (see details in **Supplementary Note 7**, also shown below). Since this intensity decays exponentially with respect to sample thickness, the penetration of MIR light through the 0.9-mm thick sample in the original manuscript is rather low. Therefore, we prepared thinner pellets of 0.4 mm to ensure effective MIR penetration. In the revision, **Fig. 2–5** in the manuscript present the updated results from the thinner pellets.

Changes in the Supplementary Information:

Page S16:

Supplementary Note 7: Thickness-dependent MIR intensity distribution in the samples

To characterize MIR penetration depth, transmitted MIR intensity across protonated samples (0.3–1.1 mm in thickness) was measured using the MIR light source in this work with a HgCdTe (MCT) MIR detector (Healthy Photon HPPD-B-D-04-10; response wavelength 2–4 μm) (Supplementary Fig. 17a). Detector output voltage (u_{MCT}) – proportional to transmitted MIR intensity – was recorded and background-corrected.²⁵ As shown in Supplementary Fig. 17b, the thickness-dependent attenuation of u_{MCT} follows the Beer-Lambert law, demonstrating measurable MIR penetration through all tested samples.²⁰ Notably, the 0.4-mm sample exhibited $\sim 1.5\times$ higher transmitted intensity than the 0.9-mm sample.

Supplementary Fig. 17. a Schematic illustration for the setup measuring thickness-

dependent MIR intensity across the protonated samples. **b** Thickness-dependent attenuation of detector output voltage u_{MCT} (proportional to transmitted MIR intensity I)²⁵ follows the Beer-Lambert law,²⁰ demonstrating measurable MIR penetration through all tested samples.

Changes in the manuscript:

Page 5:

Polycrystalline BZY pellets (inset of Fig. 1c) with a thickness of **0.4 mm** were prepared by the solid-state reaction method.^{27,50} **The pellet thickness was chosen to ensure effective MIR penetration (Supplementary Fig. 17).**

Page 7:

The enhancement ratios of bulk and GB conductivities, denoted as $(\Delta\sigma/\sigma)_{\text{Bulk}}$ and $(\Delta\sigma/\sigma)_{\text{GB}}$, are as high as **36.8%** and **53.0%** (Fig. 2f).

Fig. 2 | Characteristics of O–H stretch vibration and proton conduction with and without MIR irradiation. a Schematics of the electrical conductivity measurement setup. The MIR radiation covers the wavelength range of O–H stretch vibration. **b** Vibrational profile of O–H stretch vibration band ($\nu_{\text{O-H}}$) characterized by DRIFTS (red). The orange shadow shows the typical emission spectrum of the CW MIR light. **c** Typical EIS spectra at

160 °C in wet N₂ with (red) and without (blue) MIR irradiation at an effective MIR power density (p) of 195.2 mW cm⁻². The insets illustrate the magnification of bulk features near 150 kHz, and the equivalent circuit used for fitting. The reversibility of proton conductivity enhancement is represented by single-frequency Z' at d 150 kHz for bulk (brown) and e 1 kHz for GB (turquoise). The yellow shadows highlight the time intervals for MIR irradiation. The red and grey shadows indicate stable values of Z' and their error bands with and without irradiation. f Comparison of bulk and GB proton conductivities deconvoluted from c and corresponding enhancement ratio of conductivity ($\Delta\sigma/\sigma$).

Page 9:

The observed $\Delta\sigma/\sigma$ for both bulk and GB due to MIR irradiation are about 2 times greater than the estimated results due to the IR heating effect.

Fig. 3 | Enhancement in proton conductivity versus MIR irradiation intensity. $\Delta\sigma/\sigma$ for **a** bulk (orange) and **b** GB (turquoise) versus p controlled by adjusting the distance between the sample and MIR light source (d). The sample was maintained at 160 °C. Shaded areas in **a** and **b** illustrate the 95% confidence interval of the fitted curves. Comparison of $\Delta\sigma/\sigma$ for **c** bulk and **d** GB observed in the experiments (closed circles) and estimated with the radiative heating effect (open circles connected with dashed line).

Upon MIR irradiation, the E_a for bulk proton conduction ($E_{a,\text{Bulk}}$) decreased from 0.450 ± 0.006 eV to 0.409 ± 0.007 eV, with a statistically significant difference of approximately 0.04 eV.

Conversely, the E_a for GB proton conduction ($E_{a,\text{GB}}$) remains unchanged within the error (0.716 ± 0.016 eV and 0.698 ± 0.011 eV with and without MIR irradiation, respectively).

Fig. 4 | Temperature dependence of the proton conductivity and corresponding parameters with and without MIR irradiation. Arrhenius plots of **a** proton conductivity (σT) and **b** hopping frequency (ν). **c** Vibrational factor ($Q = \nu_0 \exp(\Delta S/k_B)$) of bulk (circle) and GB (triangle) measured with (red) and without (blue) MIR irradiation at 130–200 °C. The calculated Q and their errors are represented by dashed lines and shadowed areas. Comparison of the corresponding E_a (orange), σ_0 (blue), and Q (green) of **d** bulk and **e** GB.

The error bars contain the fitting error, as well as the error and standard deviation of three measurements for each sample, and multiple samples prepared using the same method.

Comment 3 of Reviewer #2:

Impedance:

a. The authors state that the use of their “three serial R(CPE) networks” indicate proton conduction. No, these EIS are fit with an equivalent circuit assuming proton conduction. Phrasing is important.

b. The impedance of the GB is quite broad and not well resolved. My concern is that any change in process may have led to inductivity – which would ultimately move the bulk process to lower measured resistances and only suggesting faster transport. What are the capacitances of the process, what are the ideality factors of the bulk process. Have these changed?

c. How reliable is the change in bulk transport? since the processes are not so well resolved, can the change in GB transport just have affected any bulk response? What does a DRT analysis suggest? The change in GB resistance is much higher than the bulk, ultimately leading to convoluted impedance.

d. Considering that the electrode process is not well resolved, changes in the electrode would affect the GB process here as well. A surface enhancement of transport has not been ruled out here.

Author reply: We are grateful to your valuable suggestions regarding the impedance analysis. Please see our point-to-point response below:

a. We have changed the phrasing on page 6:

The EIS spectra in Fig. 2c were then fitted to an equivalent circuit consisting of three serial R(CPE) **elements**, **assuming** proton conduction as the origin of the impedance.

b. to d. In the revision, we combined the equivalent circuit model (ECM) with DRT analysis to better resolve the bulk, GB, and electrode processes. The details are reported in **Supplementary Note 5**, also shown below. Then, we report the capacitance and ideality factors for the bulk, GB, and electrolyte processes in **Supplementary Tables 6-8**. The ideality factors for the bulk and GB processes are close to 1. The changes in capacitance and ideality factors for the bulk process upon MIR irradiation are within their errors, suggesting

that the changes result from faster bulk transport, but not changes in inductivity. We have further measured the inductive reactance of the wires, as shown in **Supplementary Fig. 18**. The inductive reactance is small compared to the bulk response, suggesting that inductance error can be ignored. The above discussion is included in **Supplementary Note 5**.

d. Please see our response to your Comment 1a for an evaluation of the surface enhancement effect.

Changes in the manuscript:

Page 23:

A combination of equivalent circuit model (ECM) and distribution of relaxation times (DRT) methods was employed to extract the parameters for bulk, grain boundary and electrode polarization processes with reduced fitting errors (Supplementary Note 5).^{82,83}

Changes in Supplementary Information:

Page S11-S13:

Supplementary Note 5: EIS spectra analysis combining equivalent circuit model (ECM) and distribution of relaxation times (DRT)

In this work, bulk and GB proton conduction, and electrode polarization processes are not well resolved in the EIS spectra. These overlaps arise when the relaxation times (τ) of the processes differ minimally.^{12,13} The fitting errors are considerable (> 60%) while determining the component parameters in such EIS spectra using the equivalent circuit model (ECM) method. On the contrary, the distribution of relaxation times (DRT) method resolves the EIS spectra into continuous curves with distinct peaks, even when the relaxation times of different processes are close.^{12,13}

To reduce fitting errors, a combination of ECM and DRT analysis was employed.¹⁴ DRT analysis was performed via DRTtools in MATLAB.¹⁵ Multiple electrochemical processes were identified according to the dominant peaks, and simulated by several R(CPE) elements. The corresponding resistances were estimated by integrating the distribution function over the $\ln\tau$ axis for each peak, and inserted as initial values of different elements in the ECM.

As presented in Supplementary Fig. 14, three distinct peaks were deconvoluted from the EIS spectra measured at 160 °C, both with and without MIR irradiation. The two peaks covering 10^5 – 10^7 Hz and 10^3 – 10^5 Hz were assigned to bulk and GB proton conduction processes, respectively.^{13,14,16} The distinct peak below 10^3 Hz, accompanied by several low-intensity, closely spaced hidden peaks, was collectively attributed to electrode polarization and modeled using a single R(CPE) element.¹⁴ Thus, we adopted the equivalent circuit consisting of three serial R(CPE) elements (Supplementary Fig. 14), assuming proton conduction as the origin of the impedance. Each R(CPE) element represents the contribution from the bulk, GB, and electrode to the overall impedance. Supplementary Tables 8–10 show the EIS fitting results, including resistance, capacitance factor, and ideality factor for the bulk, GB, and electrolyte processes. The ideality factors for the bulk and GB processes are close to 1. The changes in capacitance factors and ideality factors for the bulk process upon MIR irradiation are within their errors, suggesting that the changes result from faster bulk transport, but not changes in inductivity. Furthermore, the measured inductance of the wires, as shown in Supplementary Fig. 15, is small and negligible compared to the bulk response, suggesting inductance error¹⁷ contributes negligibly to the EIS measurements.

Supplementary Fig. 14. Assignment of different components in EIS spectra (top panel) using the DRT function (bottom panel).¹⁵ The resulting equivalent circuit model (ECM) is presented in the middle panel. The data are measured on the 0.4-mm thick sample at 160 °C in wet N₂.

Supplementary Fig. 15. Room-temperature impedance spectrum (2 MHz to 10 Hz) of the test system shorted at the sample position reveals a maximum inductive reactance (positive Z'') originating from the 2-m long leads.¹⁷ This inductive reactance represents less than 0.07% of the total measured Z'' with actual samples at 130–200 °C. Since the leads are kept at room temperature, this result demonstrates that inductance error contributes negligibly to the EIS measurements.

Supplementary Table 8. Fitting parameters of the R(CPE) circuit element for bulk proton conduction: Resistance (R_{Bulk}), capacitance factor (Y_{Bulk}), and ideality factor (n_{Bulk}). Corresponding equivalent circuit model shown in Supplementary Fig. 14.

MIR status	R_{Bulk} (Ω)	Y_{Bulk} (F)	n_{Bulk}
Off	$(1.61 \pm 0.02) \times 10^5$	$(2.26 \pm 0.42) \times 10^{-12}$	1.000 ± 0.015
On	$(1.18 \pm 0.03) \times 10^5$	$(2.20 \pm 0.27) \times 10^{-12}$	0.999 ± 0.009

Supplementary Table 9. Fitting parameters of the R(CPE) circuit element for GB conduction: Resistance (R_{GB}), capacitance factor (Y_{GB}), and ideality factor (n_{GB}). Corresponding equivalent circuit model shown in Supplementary Fig. 14.

MIR status	R_{GB} (Ω)	Y_{GB} (F)	n_{GB}
Off	$(8.75 \pm 1.72) \times 10^5$	$(2.70 \pm 0.31) \times 10^{-10}$	0.912 ± 0.038
On	$(5.72 \pm 0.93) \times 10^5$	$(2.52 \pm 0.49) \times 10^{-10}$	0.935 ± 0.031

Supplementary Table 10. Fitting parameters of the R(CPE) circuit element for electrode processes: Resistance ($R_{\text{Electrode}}$), capacitance factor ($Y_{\text{Electrode}}$), and ideality factor ($n_{\text{Electrode}}$). Corresponding equivalent circuit model shown in Supplementary Fig. 14.

MIR status	$R_{\text{Electrode}} (\Omega)$	$Y_{\text{Electrode}} (\text{F})$	$n_{\text{Electrode}}$
Off	$(5.69 \pm 0.25) \times 10^6$	$(5.65 \pm 1.09) \times 10^{-9}$	0.545 ± 0.013
On	$(3.67 \pm 0.07) \times 10^6$	$(4.23 \pm 0.53) \times 10^{-9}$	0.574 ± 0.008

Supplementary Fig. 16. Distribution of relaxation times (DRT) curves¹⁵ for bulk and GB **a** without and **b** with MIR irradiation obtained from the EIS spectra of the samples in 130–200 °C. Considering the inverse relation between the relaxation time and hopping frequency,^{14,18} the overall trend of peak position in the DRT spectra well reflects the evolution of jump frequency presented in Figure 3b of the main text.

Comment 4 of Reviewer #2:

The authors check if T increase from IR heating would affect it and conclude that the effect is 2-3 times larger. How significant is that and does it neglect thermal expansion of the material?

Author reply: We measured the temperature increase upon switching on the MIR light source while deactivating the PID feedback. The sample temperature rise is about 5 °C. Within this temperature range (5 °C above the reference sample temperature), we further calculated the changes in volume and geometric factor for conductivity calculation of BZY, which are caused by thermal expansion. These effects of thermal expansion are **less than 0.1% and negligible in the calculation of enhancement effect.**

Changes in the Supplementary Information:

Page S19:

Supplementary Note 8: Impact of IR heating effect on sample temperature

The heating effect of MIR irradiation on the samples is manifested as the percent change in conductivity ($\Delta\sigma/\sigma$) due to the temperature rise (ΔT_h) upon irradiation of light. ΔT_h is defined by the difference between: (i) the steady-state sample temperature during MIR irradiation while deactivating the PID control, and (ii) the reference temperature T . With the PID feedback loop deactivated, the thermostat's output power was fixed at its baseline level (without MIR irradiation). In this configuration, all observed sample temperature variations can be reasonably attributed to IR heating effect. As plotted in Supplementary Fig. 18, ΔT_h was found to be approximately 5 °C when T is 160 °C. The resulting $(\Delta\sigma/\sigma)_{\text{Bulk}}$ and $(\Delta\sigma/\sigma)_{\text{GB}}$ were estimated to be 15% and 25%, respectively, which are lower than those observed in experiment (36.8% and 53.0%). On the other hand, achieving the observed $\Delta\sigma/\sigma$ values solely by heating would require $\Delta T_h = 13$ °C. Such contrast demonstrates a higher energy efficiency of MIR irradiation, achieving a comparable enhancement ratio in proton conductivity at significantly lower temperatures.

To quantify ΔT_h as a function of working distance (d), we modeled the sample as a thermal resistor considering only conductive heat transfer. We further assumed complete conversion of incident optical power to heat flow (P_h), thus having $\Delta T_h \propto P_h$.²² The thickness-dependent P_h was calculated by scaling the normalized function $p(d)$ (Supplementary Fig. 10b) to the range [0,1]. As shown in Figures 3c, d of the main text, the observed MIR-induced enhancement in proton conductivity ($\Delta\sigma/\sigma$) for both bulk and grain boundaries exceed the heating-only estimates by a factor of 2–3. This significant discrepancy demonstrates that IR heating effect cannot account for the majority of the enhancement in proton conductivity.

Supplementary Fig. 18. Representative sample temperature curve upon switching on the MIR light source and deactivated PID feedback loop. The temperature rise (ΔT_h) of the sample at equilibrium is approximately 5 °C when the reference temperature is 160 °C. The resulting $(\Delta\sigma/\sigma)_{\text{Bulk}}$ and $(\Delta\sigma/\sigma)_{\text{GB}}$ were estimated to be 15% and 25%, which are lower than those observed in experiment (36.8% and 53.0%).

Supplementary Note 9: Impact of thermal expansion on sample geometry

In protonic ceramic electrochemical cells (PCECs), thermal expansion coefficient mismatch between electrode and electrolyte materials generates thermal stress under operating conditions. Such stress can initiate microcracks in the cell stack, ultimately leading to significant performance degradation.²⁸ The impact of thermal expansion with and without MIR irradiation was then evaluated.

To the best of our knowledge, there are no studies reporting material expansion due to optical effects of infrared light. Therefore, we attribute the thermal expansion solely to IR heating effects. The thermal expansion coefficient (α) of BZY20 has been reported to be $8.2 \times 10^{-6} \text{ K}^{-1}$.²⁹ For an isotropic material with initial volume V_0 at the reference temperature, the thermally expanded volume due to temperature rise ΔT_h is given by $V_T = V_0 (1 + \alpha \Delta T_h)^3$, yielding the volume ratio $V_T/V_0 = (1 + \alpha \Delta T_h)^3$.²⁸

On the other hand, the proton conductivity σ is expressed as

$$\sigma = \frac{1}{R} \frac{d}{A} = \frac{1}{R} F \quad (\text{S5})$$

where R , d , and A denote the sample's resistance, length and cross-sectional area, and $F = d/A$ is defined as a geometric factor. By analogy, the geometric factor ratio induced by ΔT_h , $F_T/F_0 = 1/(1 + \alpha \Delta T_h)$ can be obtained.

As discussed in Supplementary Section 8, ΔT_h is approximately 5 °C when the reference temperature is 160 °C. The corresponding changes in V_T/V_0 and F_T/F_0 are both less than 0.001 (Supplementary Fig. 19), confirming that thermal expansion effects are negligible

under MIR irradiation. Moreover, 5 °C is lower than the ΔT_h required to achieve the observed $\Delta\sigma/\sigma$ values only through heating (13 °C), implying significantly reduced thermal expansion, and hence thermal stress. These findings validate two key conclusions: (i) Eq. (1) in the main text is valid as the change in sample geometry is within 0.1% during MIR irradiation, and (ii) MIR irradiation would achieve significant proton conductivity enhancement while inducing minimal thermal stress in PCECs – a crucial advantage over conventional heating approach.

Supplementary Fig. 19. **a** Volume ratio (V_T/V_0) and **b** geometric factor ratio (F_T/F_0) for calculating proton conductivity showing marginal changes due to IR heating effect.

Comment 5 of Reviewer #2:

Have the authors considered that the light affects the defect formation enthalpy rather than the actual diffusion thermodynamics of the process? Basically, is it the migration enthalpy that is affected or the defect formation enthalpy?

Author reply: Thank you for proposing this possibility. Our assumption of this work was that the enhancement caused by MIR light originates from the excitation of O–H stretch vibration, thus affecting the proton mobility. This mechanism has been proved experimentally in refs. 35–36 of the manuscript.^{1,2} In proton-conducting metal oxides, defect formation enthalpy is referred to as hydration enthalpy, or proton incorporation enthalpy. Changes in hydration enthalpy will lead to changes in proton concentration. Following your suggestions, we found computational evidence that phonons can influence the hydration entropy of acceptor-doped BaZrO₃ (refs. 67–68 of the manuscript).^{3,4} The change in hydration entropy may affect the proton concentration. However, these works discuss lattice phonons, not O–H stretch vibration. Additionally, we have measured the impedance spectra of the protonated sample in wet N₂, ambient air, and dry N₂. The results have been displayed in **Supplementary Fig. 5**, also shown in our response to your Comment 1a. For the protonated

sample, the impact of both atmospheric composition and water partial pressure are not significant at 160 °C, suggesting that the hydration/dehydration processes are very slow at this temperature. **Since the enhancement occurs almost instantaneously, we believe that our observation is not due to light-induced effects on the defect formation enthalpy.**

Changes in the manuscript:

Page 15:

While the aforementioned mechanisms may increase proton mobility, higher proton concentration represents another potential origin of the enhanced proton conductivity. Computational evidence has demonstrated that phonons can influence hydration entropy of acceptor-doped BaZrO₃,^{68,69} thus changing the proton concentration. However, these investigations focused specifically on lattice phonons rather than the O–H stretching vibration central to this work. Furthermore, as shown in Supplementary Fig. 5, the variation in conductivity of the protonated sample was not significant with changing water partial pressure at 160 °C, indicating kinetically limited hydration/dehydration processes at this temperature. On the contrary, the observed MIR-induced conductivity enhancement occurs instantaneously below 200 °C. Given this rapid response, we believe that our observation is not dominated by the light-induced effects on hydration enthalpy and the subsequently increased proton concentration.

Comment 6 of Reviewer #2:

There seems to be a misconception about the apex frequency in the impedance. On page 10, the authors use the apex frequency of the bulk process ($\omega = 1/RC$) and suggest that this is the jump frequency ν . These are two entirely different parameters.

Author reply: Thank you for pointing out this misconception. Indeed, equating jump frequency with the apex frequency of the bulk process ($1/RC$) is misleading. They are distinct physical quantities operating at different scales. We changed the expression for ω ($1/RC$) to "characteristic frequency of the bulk process".

Changes in the manuscript:

Page 10:

In practice, the variation trend of ν upon MIR irradiation can be evaluated from the fitting results of EIS spectra. ... Subsequently, we can obtain the characteristic frequency for bulk or

GB processes, ω , using $\omega = 1/(RC)$ a vibrational factor $Q = \omega_0 \exp(\Delta S/k_B) = \omega \exp(E_a/k_B T)$, as derived from Eq. (3).

Page 11:

Although ω and ω_0 are obtained from EIS and represents macroscopic measurements, they can reflect the changes in microscopic proton hopping frequency ν and attempt frequency ν_0 , respectively.^{49,60,61}

Comment 7 of Reviewer #2:

Figure 4 a gives the bulk and grain boundary conductivities. Grain boundary conductivities cannot be calculated from the impedance data without the in-depth knowledge of the microstructure. See J Power Source 2011, 196, 6456. If these information are not known, one can only plot the resistances but not the conductivity. Arrhenius plots and activation barriers can be generated from the inverse resistance.

Author reply: Thank you for pointing out this important issue. We calculated the specific grain boundary (GB) conductivity derived from the brick-layer model⁵ to account for the microstructure, and adopt this value as the “GB conductivity” throughout this work. The results are presented in the revised Fig. 4. Please also refer to the relevant highlighted changes in Supplementary Note 6.

Changes in the manuscript:

Page 7:

The GB conductivity (σ_{GB}) is defined as the specific value derived from the brick-layer model (Supplementary Note 6) to account for the microstructure.^{52,53}

Changes in the Supplementary Information:

Page S16:

Supplementary Note 6: Model for the grain boundary (GB) conductivity

The brick-layer model was employed to include the information of microstructure while evaluating grain boundary (GB) conductivity.^{19,20} It treats the real microstructure of the samples (Supplementary Fig. 2) as an array of cube-shaped grains separated by flat GB. This model gives two available paths for current conduction: (i) through grains and across GB; (ii)

along GB. In BZY20 pellets, proton conduction following path (ii) dominates, where bulk and GB are connected in series. The specific GB conductivity (termed GB conductivity and σ_{GB} throughout this work) considers the microstructure through the ratio of grain boundary thickness (g) to the grain size (G):^{19,21}

$$\sigma_{\text{GB}} = \frac{1}{R_{\text{GB}}} \frac{d}{A} \frac{g}{G} \quad (\text{S1})$$

where R_{GB} is the apparent GB resistance fitted from EIS spectra, and d and A are the length and cross-sectional area of the sample, respectively. Assuming the dielectric constant is about the same for the bulk and the GB, σ_{GB} can be calculated without microstructure examination:^{19,21}

$$\sigma_{\text{GB}} = \frac{1}{R_{\text{GB}}} \frac{d}{A} \frac{C_{\text{Bulk}}}{C_{\text{GB}}} \quad (\text{S2})$$

where C_{Bulk} and C_{GB} are the pseudocapacitance of bulk and GB, respectively.

Comment 8 of Reviewer #2:

Should the proposed mechanism not lead to a change in activation barrier? But this is experimentally not observed, the proposed mechanism would suggest it.

Author reply: In the 0.4-mm thick sample, we observe a higher enhancement ratio than the 0.9-mm pellet; however, the change in activation energy is still small (about 0.04 eV). Therefore, we made a correction to the proposed mechanism, as shown in Figure 5 and the relevant text.

Changes in the manuscript:

Abstract:

We rationalize the enhancement as the excitation of O–H stretch vibrational states, followed by the relaxation into lattice vibration modes, modulating the potential energy surface of the proton.

Page 16:

The resonant photon energy excites the vibrational energy to a higher vibrational state (Fig. 5c and Supplementary Fig. 20a), and promotes proton transfer in the lattice.

Fig. 5 | Wavelength effect and suggested mechanism of MIR light enhanced proton conductivity. **a** Representation of passing wavelength ranges of the MIR narrowband bandpass filters A (brown shadow) and B (green shadow). The range for filter A is closest to the resonant frequency of O–H stretch vibration. **b** Wavelength effect of MIR light enhanced $\Delta\sigma/\sigma$ of bulk and GB for filters A and B, and the control group. **Schematic representations for the lattice vibration-assisted proton hopping process showing c excitation and relaxation of the O–H stretch vibration, d modulation of the PES of the proton due to subsequent excitation of the coupled lattice vibrations, and e proton transfer from the donor (O_I) to the acceptor (O_{II}).** For reference, the dashed curve in **d** depicts the shape of the PES prior to resonant excitation, corresponding to configuration shown in **c**.

Page 18:

However, if proton transfer occurs via direct breaking of the O–H bond when the proton is at the first excited state, it would lead to a significant decrease in the activation energy, deviating from our observation that the change in $E_{a,Bulk}$ upon MIR irradiation is only 0.04 eV. Therefore, we propose that the enhanced proton conduction originates from relaxation of

excited O–H stretch vibration into lower-frequency oxygen lattice vibrations coupling to O–H stretch vibration (Fig. 5c). These coupled vibrations alter the O–O distance, thereby tuning the height and width of the PES of the proton (Fig. 5d), and facilitating proton transfer (Fig. 5e).^{14,72,73} For proton-conducting metal oxides, experimental evidence has confirmed that the temperature-dependent excited-state lifetimes of O–H stretch mode of KTaO₃ is coupled to the O–Ta–O bending motion, thus reducing the activation barrier height.^{35,67,74} Our previous works have also shown that the temperature-dependent proton jump times of BZY and Y-doped BaCeO₃ measured by quasielastic neutron scattering follow Samgin's proton polaron model,^{75–77} suggesting that proton hopping is strongly coupled to lattice phonons, in particular B–O stretch modes.^{28,46} The above evidence indicates that the lattice vibrations coupled to O–H stretch vibration are able to tune the PES of the proton, and, in turn, assist proton transfer.

Comment 9 of Reviewer #2:

What do the authors mean with “The fluctuation of the configurational entropy may be amplified”. What is a fluctuation of a configurational entropy in such a disordered proton conductor?

Author reply: Our original intension is only to emphasize the "change" in the configurational entropy. We have changed the wording.

Changes in the manuscript:

Page 14:

The **change in** configurational entropy **during proton conduction** may be amplified by the excitation of lattice vibrations coupled with O–H stretch vibration...

Reviewer #3

Comments of Reviewer #3:

It is highly meaningful to find and establish some efficient approaches to improve ionic conduction. As stated by the authors, MIR-induced enhancement of proton conduction has been reported bur scarce (refs 41 and 42). This work demonstrates experimentally that this method is quite effective for proton conductor. I think this article is suitable for publication in

Nat Commun after careful clarification.

Author reply: Thank you for the encouraging remarks and useful suggestions. We address all comments one by one below. All modifications have been highlighted in the revised manuscript.

Comment 1 of Reviewer #3:

The test was conducted in ambient air. As reported by many reports, the air humidity has a large impact on proton conductivities. This impact may be more remarkable when the MIR light is on. A valuation in N₂ or vacuum is suggested.

Author reply: In the revised manuscript, we performed the experiments again on protonated samples in H₂O-saturated N₂ (wet N₂) in a gas-tight chamber. A photo of the measurement setup in the chamber is shown in **Supplementary Fig. 10**. The revised results are presented in **Fig. 2–5** and the relevant text. Please also refer to the highlighted changes in the revised manuscript.

Additionally, we have also investigated the effects of atmosphere by comparing the impedance results in wet N₂, ambient air and dry N₂. The results show that the impact of atmosphere is not significant at 160 °C, see **Supplementary Note 2**.

Changes in the manuscript:

Page 5:

Their electrical conductivities were measured by EIS with and without MIR irradiation in H₂O-saturated N₂ (wet N₂; $p(\text{H}_2\text{O}) = 0.02 \text{ atm}$) between 130 and 200 °C.

Page 22:

All measurements were performed in a gas-tight chamber supplied with ambient air, dry N₂ ($p(\text{H}_2\text{O}) < 10^{-4} \text{ atm}$), H₂O-saturated N₂ (wet N₂; $p(\text{H}_2\text{O}) = 0.02 \text{ atm}$), and D₂O-saturated N₂ ($p(\text{D}_2\text{O}) = 0.02 \text{ atm}$), respectively. Prior to measurements in each atmosphere, the samples were conditioned in the target atmosphere for over 1 h.

Changes in the Supplementary Information:

Page S6:

Supplementary Fig. 5. Nyquist plots of protonated BZY samples measured at 160 °C in H₂O-saturated (wet) N₂, ambient air, and dry N₂ with MIR irradiation **a** off and **b** on. The insets illustrate the magnification of bulk features. The atmosphere in the gas-tight chamber was switched from H₂O-saturated N₂ to ambient air, then to dry N₂. The target atmosphere was purged into the chamber for over 1 h at a high gas flow rate before impedance measurements were taken. No significant changes were observed in the impedance spectra in different atmospheres, suggesting no significant hydration/dehydration processes at 160 °C.

Page S8:

Supplementary Note 3: Additional information about the lab-made test system

Supplementary Fig. 8. Schematic illustration and optical image for setups **a** without and **b**

with the IR bandpass filter of the lab-made test system. d denotes working distance (the distance between top surface of the sample and emission window of the light source). The bandpass filter is in contact with sample fixture and the light source. **c Optical image of the gas-tight chamber enclosing the lab-made test system with internal dimensions of 15 cm × 10 cm × 20 cm. Different atmospheres were obtained by purging the chamber with the target atmospheres for over 1 h.**

Comment 2 of Reviewer #3:

I am very curious about the homogeneity of protons in the material and its effect on MIR-induced enhancement of the proton conductivities. The protonated BZY was prepared as “Protonation of the as-sintered pellets was performed in wet nitrogen at 600 °C for 24 h at a ramp rate of 5 °C min⁻¹”. It is possible that the concentration of the proton atom decreases from the surface to the interior. As shown in the test setup (Fig. 2a), the two Ag electrodes were placed directly on the upper surface of the pellet. The conduction path on the surface would be dominant supposing the higher H concentration on the surface.

Author reply: To find out whether the proton concentration is higher on the surface, we protonated a pellet of about 1.1 mm thick using the same method. Then we broke the pellet and used Raman microscopy to map the intensity of O–H stretch peak at different positions on the cross section. The result may provide an idea about the **proton distribution across the pellet thickness**. As presented in **Supplementary Fig. 4**, the O–H stretch intensity varies at different positions, due to the scattering of laser light by different orientations of grains. However, we did not observe higher proton concentration (very high O–H stretch peak intensity) on the surface. Since the proton conductivity were measured from a thinner pellet of 0.4 mm, it is reasonable to assume that the proton concentration at the interior of the pellet is similar to that at the surface. Note that it is not possible to acquire Raman spectra on the pellet edge as it is not flat and hard to focus.

Changes in the manuscript:

Page 20:

Supplementary Fig. 4 presents the Raman intensity depth profile of O–H stretching band on the cross-section of a protonated BZY pellet, suggesting that the proton concentration at the interior of the pellet is similar compared to that at the surface.

Changes in the Supplementary Information:

Page S5:

Supplementary Fig. 4. Raman intensity depth profile of O–H stretching band on the cross-section of a protonated BZY pellet of 1.1 mm-thick. A single peak from molecular N₂ in the air appears near 2331 cm⁻¹.^{3,4}

Comment 3 of Reviewer #3:

The authors tried to exclude the impact of MIR-induced thermal effect. The monitoring of the pellet temperature was on the lower surface of the material. Likewise, as shown in Fig. 2a, the MIR light was directly shined to the upper side of the material and the two electrodes. In this case, the thermal diffusion from the upper side to the bottom may be not so rapid as we imagine. So, the thermal effect should be carefully re-valuation. Test the temperature of the upper surface?

Author reply: In the revision, we had to **measure the pellet temperature from both upper (top) and bottom sides using two thermocouples**. After the MIR light was switched on for about 3 min, the top and bottom temperatures were almost the same. Additionally, we have measured the thermal conductivity of the BZY, and calculated the theoretical time taken for the pellet to reach thermal equilibrium, which is about 75 s. However, because of better environmental stability of temperature measurement, we decide to monitor the sample temperature from the bottom because the top surface temperature measured by the thermocouple is reported to be sensitive to ambient conditions and direct MIR exposure.⁶

Changes in the Supplementary Information:

Page S9:

Two thermocouples were mounted to the top and bottom surfaces of the pellet to monitor temperature (T), respectively. Both thermocouples showed consistent readings after ~3 min of MIR irradiation (Supplementary Table 4). We chose the bottom thermocouple for T measurements because of better environmental stability (Supplementary Fig. 11), as the top thermocouple was more sensitive to ambient conditions and direct MIR exposure.¹⁰ This selection is further justified by thermal conduction analysis. The thermal conduction time constant ($\tau_h = C_p h / (\lambda A)$),¹¹ determined using the sample thickness ($h = 0.4$ mm), MIR spot area ($A = 7$ mm²), and measured thermal properties (Netzsch LFA 467) (specific heat capacity $C_p = 0.46$ J g⁻¹ K⁻¹; thermal conductivity $\lambda = 2.1$ W m⁻¹ K⁻¹), yields $\tau_h \approx 15$ s. This indicates that thermal equilibrium is effectively achieved within $5\tau_h$ (75 s).¹¹ Our EIS measurements were performed after 3 min of MIR irradiation, ensuring stable temperature distribution in the sample, confirming the appropriateness of our thermocouple configuration.

Page S10:

Supplementary Table 4. Readings of thermocouples mounted to the sample's top and bottom surfaces after ~3 min of MIR irradiation.

T_{Top} (°C)	T_{Bottom} (°C)
160.8	160.7

Comment 4 of Reviewer #3:

The symbol “GB” should be noted when its full name first appears.

Author reply: We clarified the full name of GB when it first appeared.

Changes in the manuscript: Page 4:

We demonstrate reversible switching between high and low bulk and **grain boundary (GB)** resistances controlled by MIR irradiation measured by electrochemical impedance spectroscopy (EIS) while maintaining the sample temperature constant.

Comment 5 of Reviewer #3:

I am not an expert on the theoretical aspect described in the manuscript, which requires

valuation by other referees.

Author reply: We have revised the proposed theoretical model according to our revised results and **Comment 8, Reviewer #2**.

We thank all the reviewers again for their valuable comments.

References

1. Spahr, E. J. *et al.* Proton tunneling: a decay channel of the O–H stretch mode in KTaO_3 . *Phys. Rev. Lett.* **102**, 075506 (2009).
2. Spahr, E. J. *et al.* Giant enhancement of hydrogen transport in rutile TiO_2 at low temperatures. *Phys. Rev. Lett.* **104**, 205901 (2010).
3. Bjørheim, T. S., Løken, A. & Haugsrud, R. On the relationship between chemical expansion and hydration thermodynamics of proton conducting perovskites. *J. Mater. Chem. A* **4**, 5917–5924 (2016).
4. Bjørheim, T. S., Kotomin, E. A. & Maier, J. Hydration entropy of BaZrO_3 from first principles phonon calculations. *J. Mater. Chem. A* **3**, 7639–7648 (2015).
5. Haile, S. M., West, D. L. & Campbell, J. The role of microstructure and processing on the proton conducting properties of gadolinium-doped barium cerate. *J. Mater. Res.* **13**, 1576–1595 (1998).
6. Le Maout, Y. & Schmidt, F. Infrared Radiation Applied to Polymer Processes. in *Heat Transfer in Polymer Composite Materials* (ed. Boyard, N.) 385–423 (Wiley, 2016). doi:10.1002/9781119116288.ch13.